# VISUAL MULTI-AGENT SYSTEM: MITIGATING HALLUCINATION SNOWBALLING VIA VISUAL FLOW

**Xinlei Yu**[1]   **Chengming Xu**[2]   **Guibin Zhang**[1]   **Yongbo He**[3]   **Zhangquan Chen**[4]
**Zhucun Xue**[3]   **Jiangning Zhang**[3]   **Yue Liao**[1]   **Xiaobin Hu**[1*]   **Yu-Gang Jiang**[5]
**Shuicheng Yan**[1]
[1]National University of Singapore   [2]Tencent Youtu Lab   [3]Zhejiang University
[4]Tsinghua University   [5]Fudan University
`xinlei.yu@u.nus.edu`   `ben0xiaobin0hu1@nus.edu.sg`
∗ Corresponding Author

## ABSTRACT

Multi-Agent System (MAS) powered by Visual Language Models (VLMs) enables challenging tasks but suffers from a novel failure term, ***multi-agent visual hallucination snowballing***, where hallucinations are seeded in a single agent and amplified by following ones due to the over-reliance on textual flow to relay visual information. Through turn-, layer-, and token-wise attention analyses, we provide detailed insights into the essence of hallucination snowballing regarding the reduction of visual attention allocation. It leads us to identify a subset of vision tokens with a unimodal attention peak in middle layers that best preserve visual evidence but gradually diminish in deeper agent turns, resulting in the visual hallucination snowballing in MAS. Thus, we propose **ViF**, a lightweight, model-agnostic mitigation paradigm that relays inter-agent messages with **Vi**sual **F**low powered by the selected visual relay tokens and applies attention reallocation to amplify this pattern. The experiment results demonstrate that our method markedly reduces hallucination snowballing, consistently improving the performance across eight benchmarks based on four common MAS structures and ten base models. The source code is publicly available at: https://github.com/YU-deep/ViF.git.

## 1 INTRODUCTION

MAS equipped with advanced VLMs are rapidly emerging as a solution for complex tasks, such as collaborative reasoning, multi-turn instruction following, and sophisticated multi-modal understanding, by enabling agents to communicate and collaborate over multiple turns so as to tackle problems that are intractable for a single model (Cemri et al., 2025; Li et al., 2025b). However, this collaboration also exposes a fundamental reliability failure due to the problem of ***multi-agent visual hallucination snowballing***, that is, visual misinterpretations or over-preference to textual messages in previous agents that are amplified as information flows through subsequent agents, producing propagatively hallucinated outputs about the visual contents and, ultimately, catastrophic hallucination snowballing. This introduces new reliability and effectiveness challenges in VLM-based MAS that can not be addressed by single-agent research.

It is noteworthy that the visual hallucination snowballing phenomenon in MAS is essentially different from such problem discussed in the previous works (Zhang et al., 2024; Zhong et al., 2024), given that the hallucination snowballing arises from two distinct but interacting mechanisms, as shown in Figure 1a: (1) intrinsic hallucination, where individual VLM-based agent produces erroneous textual descriptions or assertions about visual contents, and (2) hallucination propagation, where the ***over-reliance on textual information flow***, *i.e*, the generated text, compresses and selectively emphasizes visual features, allowing surviving hallucinated assertions to be treated as authoritative by downstream agents. Since later agents typically accept prior textual context as strong evidence, early hallucinations are hence amplified rather than corrected, producing a snowballing effect across turns. Due to the interaction between these two mechanisms, reducing per-agent hallucination alone, as focused by previous works (Wang et al., 2025; Tang et al., 2025b; Yin et al., 2025; Li et al., 2025c; Zou et al., 2025; Tang et al., 2025a), cannot fully solve the hallucination propagation problem, thus failing to prevent multi-agent hallucination snowballing.

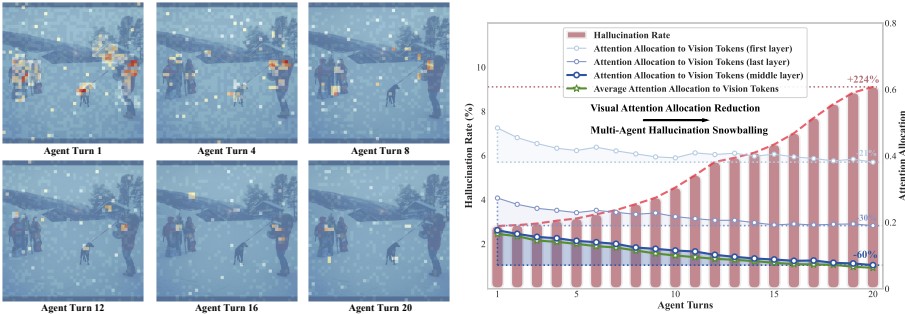

(a) An example of multi-agent hallucination snowballing, and a comparison of visual information relay between existing *i)* textual flow approaches and *ii)* our proposed visual flow method.

(b) Maps of average visual attention allocation across agent turns.

(c) Relations between visual attention allocation reduction and hallucination snowballing.

Figure 1: Introduction to the multi-agent visual hallucination snowballing phenomenon: (a) presents an example illustrating how it happens; (b) and (c) specify the visual attention allocation reduction in different agent turns, potentially contributing to the occurrence of hallucination snowballing.

To diagnose how multi-agent pipelines lose visual fidelity across turns, we first conduct a set of preliminary analyses that dissect attention dynamics among turn-wise, layer-wise, and token-wise, through which we empirically conclude that the hallucination snowballing can be evident by the reduction of attention allocated to vision tokens over agent turns, as indicated in Figure 1b and Figure 1c. Moreover, vision tokens characterized by ***unimodal attention peak in middle layers***, as a small but vital subset of all vision tokens, can best preserve vision-specific information and whose removal most degrades visual understanding, thus being significant for enhancing the visual information flow among agents. Such a token pattern, however, diminishes in deeper agent turns, implying the gradual dominance of textual information flow, leading to the hallucination snowballing.

Motivated by these insights, we propose an innovative mitigation strategy for multi-agent hallucination snowballing dubbed as ViF. Instead of relying solely on textual flows, an additional ***visual flow*** is introduced to relay visual evidence by selecting a subset of visual relay tokens and being contextualized by previous instructions, then engaging them in the process of following agents. Such a design can provide downstream agents with preserved visual evidence that resists visual-to-text information loss, meanwhile preventing textual priors from entirely displacing visual signals during subsequent agent turns. In addition, an attention reallocation mechanism is introduced to amplify the ideal attention patterns and preserve visual contributions into deeper agent turns. We evaluate ViF across eight benchmarks covering both comprehensive and hallucination tasks, demonstrating its striking effectiveness in alleviating hallucination snowballing in four different structures and ten base VLMs. Overall, our contributions are summarized as follows:

- We formalize the multi-agent visual hallucination snowballing phenomenon and systematically link it to visual attention degradation in deeper agent turns.
- We provide extensive analyses that identify a subset of vision tokens that are critical for relaying visual information flow.
- We introduce ViF, a model-agnostic method that optimizes inter-agent visual messages with visual flows and an attention reallocation mechanism to augment attention patterns.
- Comprehensive experiments validate the efficacy of our ViF to reduce hallucination snowballing, and additional analyses provide more convincing evidences.

## 2 REQUISITE ANALYSES

As mentioned in Section 1, the hallucination snowballing can be presented by the negative correlation with the attention allocation to vision tokens. Quantitatively, as shown in Fig. 1c, the average attention allocation to vision tokens reduces from 0.165 to 0.099 at the 10th agent turn, and further to 0.063 at the 20th turn, with a total 62% reduction. Furthermore, the reduction in the middle layer (-60%) is much more remarkable than that in the first (-21%) and last (-30%) layers. For more thorough understanding, we conduct extensive requisite analyses among various VLMs. For simplicity, we mainly focus on LLaVA-NeXT-7B (Liu et al., 2024b) on the POPE (Li et al., 2023) benchmark in the main paper, while Appendix B provides more detailed settings and comprehensive results on six VLMs to support the generalization ability of our claims, from which several insights are derived, thus leading to our research motivations.

### 2.1 ANALYTICAL EXPERIMENTS

**Layer-Wise Attention Allocation in Different Agent Turns.** To find out the underlying cause of visual hallucination snowballing in MAS, we begin by measuring the trend of layer-wise attention allocation among different agent turns. In VLMs with multi-modal architectures, the decoder dynamically allocates attention to three types of textual tokens (instruction, system and output tokens) and one visual token that produced by visual encoder. Other special tokens, such as start and end signals of visual input, are negligible and thus excluded.

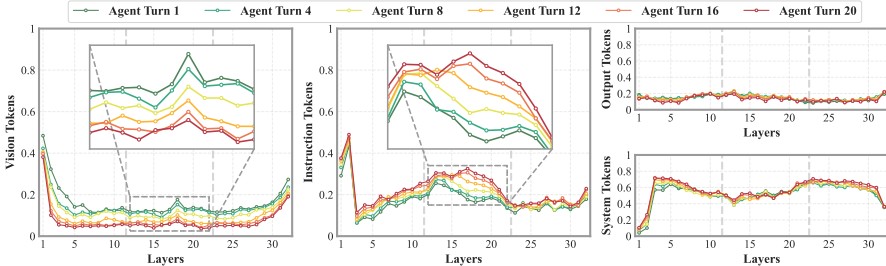

Figure 2: Layer-wise attention allocation of four tokens in different agent turns.

As depicted in Figure 2, in general, when the agent turns increase, vision tokens receive gradually decreasing attention in all layers, while the attention being directed towards instruction tokens is raised accordingly; the attention allocations to system token and output token are relatively stable, without discernible trend of change. Focusing further on the middle layers of visual and instruction tokens, which are zoomed in, the opposite trend between visual and instruction tokens is more pronounced than other layers. In the first agent turn, a scenario equivalent to a single-agent setting, there exists an obvious unimodal morphology peak in vision attention, and allocation to instruction is reduced conversely. However, in the 20th turn, the vision attention peak has almost disappeared and evolved into a fluctuation, and is redistributed to instruction tokens. Based on previous research (Yin et al., 2025; Zhang et al., 2025b) that textual and visual information are mainly fused and interacted in these layers, this tendency of attention in MAS suggests that agents in later turns tend to largely ignore the vision tokens and over-rely on the instruction tokens, including visual contents relayed by textual output from previous agents. This preference for textual tokens, however, partially leads to the multi-agent hallucination snowballing. The previous agent may experience visual-to-text information loss and potential cognitive bias when relaying visual evidence through textual flow. Conversely, vision tokens, as the initial visual semantic carrier, contain native and unbiased visual messages, which reduces the potential for hallucinations when relaying visual information. Based on these observations, we hypothesize: ***Can a subset of vision tokens, acting as visual flow, directly relay visual information across agent turns?***

**Dropping Subsets of Vision Tokens in Different Layers.** To intuitively verify the hypothesis, we ablate specific subsets of vision tokens in shallow/middle/deep layers (implementation modified from (Zhang et al., 2025a)) and compare the corresponding performance degradation. We choose five subsets of vision tokens to ablate: (1) Random Tokens: randomly select tokens from the whole vision token set and maintain a relatively uniform distribution in the image. (2) Inactive Tokens:

Table 1: Results of dropping vision token subsets in the shallow, middle, and deep layers.

| | Shallow Layers | | | | Middle Layers | | | | Deep Layers | | | |
|---|---|---|---|---|---|---|---|---|---|---|---|---|
| | 25% | 50% | 75% | 100% | 25% | 50% | 75% | 100% | 25% | 50% | 75% | 100% |
| *w/o* Dropping | 85.2 | | | | | | | | | | | |
| (a) Random | 51.8 ↓33.4 | 44.5 ↓40.7 | 38.4 ↓46.8 | 30.5 ↓54.7 | 78.9 ↓6.3 | 66.1 ↓19.1 | 62.7 ↓22.5 | 59.0 ↓26.2 | 84.4 ↓0.8 | 83.2 ↓2.0 | 82.9 ↓2.3 | 82.6 ↓2.6 |
| (b) Inactive | 55.1 ↓30.1 | 46.2 ↓39.0 | 41.8 ↓43.4 | 32.5 ↓52.7 | 84.3 ↓0.9 | 82.9 ↓2.3 | 81.5 ↓3.7 | 78.3 ↓6.9 | 85.0 ↓0.2 | 84.6 ↓0.6 | 84.3 ↓0.9 | 84.6 ↓0.6 |
| (c) Rise | 41.9 ↓43.3 | **35.6** ↓49.6 | **29.6** ↓55.6 | 20.8 ↓64.4 | 79.4 ↓5.8 | 64.2 ↓21.0 | 56.4 ↓28.8 | 52.3 ↓32.9 | **83.6** ↓1.6 | **82.7** ↓2.5 | **81.8** ↓3.4 | **81.6** ↓3.6 |
| (d) Fall | **41.6** ↓43.6 | 38.8 ↓46.4 | 30.7 ↓54.5 | 22.5 ↓62.7 | 78.3 ↓6.9 | 64.8 ↓20.4 | 58.5 ↓26.7 | 52.9 ↓32.3 | 84.1 ↓1.1 | 82.8 ↓2.4 | 82.0 ↓3.2 | 82.4 ↓2.8 |
| (e) Unimodal | 42.1 ↓43.1 | 37.6 ↓47.6 | 30.0 ↓55.2 | 22.8 ↓62.4 | **52.9** ↓32.3 | **44.5** ↓40.7 | **36.6** ↓48.6 | **25.3** ↓59.9 | 84.4 ↓0.8 | 83.0 ↓2.2 | 82.3 ↓2.9 | 81.8 ↓3.4 |

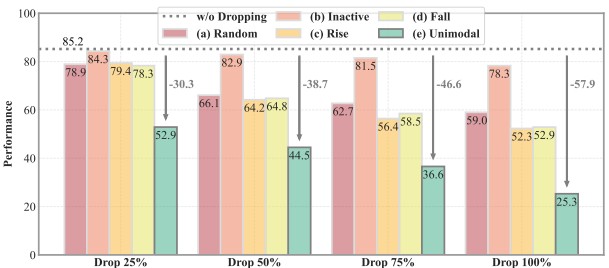

Figure 4: The demonstrations of selected unimodal vision tokens in various cases.

select the tokens with constantly low attention and tiny fluctuation. (3) Rise Tokens and (4) Fall Tokens: select tokens allocated with a gradually upward or downward trend of attention. (5) **Unimodal Tokens**: select tokens allocated with an unimodal attention peak. In terms of the unimodal tokens, we introduce a parameter $\omega$ to regulate the salience of the attention peak.

As listed in Table 1, the ablation of vision tokens leads to varying degrees of performance degradation across layers. In shallow layers, all vision tokens are necessary for the visual understanding capacity; even ablating one-quarter of the tokens from any subset causes over a 30% loss. On the contrary, in the deep layers, vision tokens play an insignificant role, with performance reduction of less than 1% even when dropping all inactive vision tokens. Figure 3 further highlights that, compared to the results of other subsets, the dropping of unimodal tokens in middle layers leads to

Figure 3: Performance when dropping different vision token subsets in middle layers.

more significant performance degradation. Specifically, the decrease from this subset is almost three times that of other subsets when dropping one-quarter of the tokens, and about twice when dropping half, three-quarters, and all tokens. In conclusion, the ablation study reveals that in shallow and deep layers, all vision tokens are almost equally important or unimportant; however, in the middle layers, vision tokens with unimodal morphology play a much more crucial role in interaction information between vision and text tokens.

**Investigation of Unimodal Tokens.** To validate our previous hypothesis that certain vision tokens can act as a visual flow for relaying visual information, we visualize the unimodal vision tokens in various cases and track their ratios across agent turns. As demonstrated in Figure 4, we choose two images, each with three distinct questions, as examples. The selected tokens are highly semantically relevant and contain very few other irrelevant tokens. Besides, as depicted in Figure 5, the proportion of unimodal vision tokens continuously declines from 1.22% at the first agent turn to 0.10% at the 20th agent turn, while the percentages of the other two tokens slightly increase. The downward trend of the unimodal token proportion aligns with visual attention allocation among agent turns, which suggests that the reduction of unimodal vision tokens contributes significantly to the onset of hallucination snowballing through the disappearance of the visual attention peak.

Thus, we believe that the subset of uni-modal vision tokens could meet the hypothesis to relay visual evidence as visual flow. It ensures that semantically relevant visual messages can be relayed suffi-

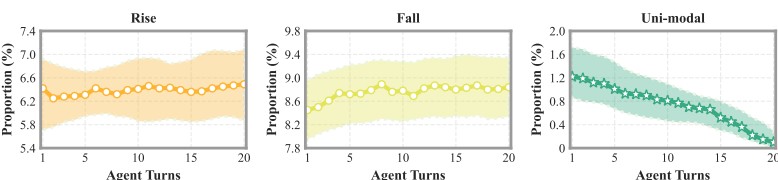

Figure 5: Proportions of vision tokens subsets in different agent turns.

ciently, while avoiding carrying substantial irrelevant vision tokens with high efficiency.

## 2.2 INSIGHTS

Based on experimental results and analyses, three significant insights can be summarized:

- The visual evidence relayed in MAS, which is typically via textual flow, potentially results in multi-agent hallucination snowballing.

- When the agent turns increase, the average attention allocated to vision tokens reduces, and the attention peak in middle layers diminishes, while attention to instruction tokens increases accordingly; system and output tokens receive relatively stable attention.

- In middle layers, vision tokens with unimodal attention allocation relay visual information; all vision tokens are significant in shallow layers and less significant in deep layers.

## 3 METHODOLOGIES

Building on the insights from the previous section, we propose a straightforward and efficient model-agnostic method named ViF to mitigate hallucination snowballing in VLM-based MAS. As demonstrated in Figure 6, our proposed method involves relaying the visual information from the previous agent via a selected subset of vision tokens, and reallocating attention in middle and deep layers to facilitate this process. Besides, we provide a suitable alternative for attention score based token selection, since in some recently released models with Flash-Attention 2/3 (Dao, 2024), the attention scores are not accessible.

### 3.1 VISUAL INFORMATION RELAY

Leveraging the previous insights, we employ the unimodal vision tokens as additional visual flow to relay information from the previous agent. Specifically, we token-wise decompose the vision tokens $\mathcal{V} = \{v_1, \ldots, v_m\}$ according to the trend of the attention allocation in the middle layers, and select the vision tokens with unimodal morphology as initial visual relay tokens $\mathcal{R} = \{r_1, \ldots, r_n\} \subset \mathcal{V}$, where $n \ll m$. However, the original selected tokens are semantically irrelevant, which are only tokenized by the vision encoder, without particular semantics. Thus, we contextualize the initial visual relay tokens $\mathcal{R}$ with the instruction tokens $\mathcal{I}$ as follows:

$$\widehat{\mathcal{R}} = f\left(\mathcal{R} \oplus \mathcal{I}\right)[:n], \tag{1}$$

where $f\left(\cdot\right)$ is a lightweight transformer block (Mehta et al., 2021), $\oplus$ denotes concatenation. Here, we extract the former $n$ component to maintain the initial length of visual relay tokens $\widehat{\mathcal{R}}$.

To retain the spatial information of visual relay tokens, we apply the same positional encoding strategy as the previous agent. Then, the visual relay tokens will be transmitted to the subsequent agent, which will be inserted between the original vision tokens and instruction tokens, and be fed to the final LLM together with other tokens.

### 3.2 ATTENTION REALLOCATION

Considering the insights that tokens are of different significance in the shallow, middle, and deep layers respectively, we reallocate attention to optimize attention patterns. Our objectives are to

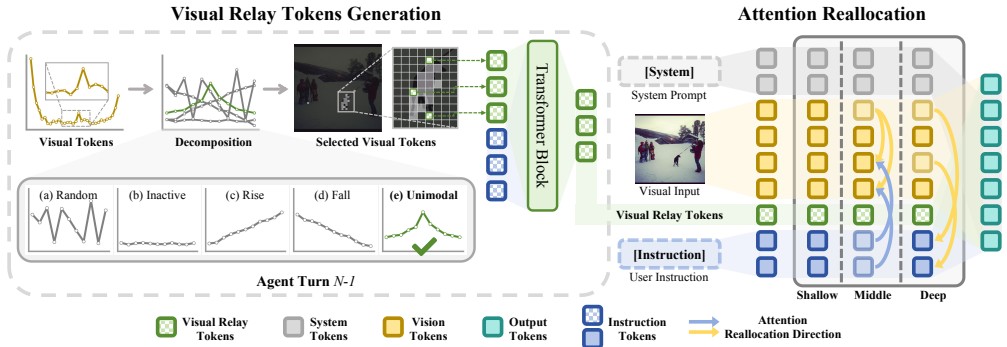

Figure 6: Overview of our proposed ViF, including the generation of visual relay tokens and attention reallocation to alleviate multi-agent hallucination snowballing.

activate the visual relay tokens and to optimize the distribution of attention among various tokens. Therefore, we amplify dynamic trends, both the upward and downward, of visual attention in the middle layers by adding temperature scaling to the Softmax operation in the middle layers:

$$\mathcal{A} = Softmax_\tau(\mathcal{S}) = \frac{\exp\left(\frac{s}{\tau}\right)}{\sum_{i=1}^{m} \exp\left(\frac{s_i}{\tau}\right)}, \tag{2}$$

where $\tau$ is temperature parameter, $\mathcal{S}$ and $s$ are attention score matrix and attention score, and $\mathcal{A}$ is attention matrix. It promotes the emergence of vision tokens with unimodal morphology. Besides, in the middle layers, we collect the attention of inactive vision tokens and instruction tokens, and then reallocate the collected attention to other vision tokens, which is formulated as:

$$\mathcal{C} = \alpha \sum_{i=1}^{m} s_i \circ M_c, M_c(i,j) = \mathbb{I}\left((i \in \mathcal{T}, j \in \mathcal{V}_\oslash) \vee (i \in \mathcal{T}, j \in \mathcal{I})\right), \tag{3}$$

$$\hat{s} = s + \frac{s}{\sum_{i=1}^{l} s_i} \mathcal{C} \circ M_r, M_r(i,j) = \mathbb{I}(i \in \mathcal{T}, j \in \mathcal{V}_\circ), \tag{4}$$

where $M_c$ and $M_r$ are the collection and reallocation mask matrices respectively, which designate the source and destination of the attention reallocation. Besides, $\alpha$ is the reallocation coefficient, $\mathcal{T}$ is the whole token set, $\mathcal{V}_\oslash \subset \mathcal{V}$ is inactive vision token set, and $\mathcal{V}_\circ = \mathcal{V} - \mathcal{V}_\oslash = \{v_1, \ldots, v_l\}$. During the reallocation, the sum of the total attention is always 1. Additionally, in the deep layers, the reallocation is from vision tokens to instruction tokens, with the same process. Thus, the two mask matrices and the reallocation coefficient are modified correspondingly.

### 3.3 ALTERNATIVE OF ATTENTION SCORE BASED STRATEGY

To accelerate the computation and reduce memory, flash-attention (Dao, 2024) mechanisms are widely used in recently released models, making the attention scores not obtainable. Inspired by (Wen et al., 2025), we design a Key-Norm ($L_2$ norm of the key matrix) based alternative for the original attention score based method. More discussions about the alternative are in Appendix C.2.

### 4 EXPERIMENTS

We conduct experiments on three comprehensive benchmarks: MME (Yin et al., 2024), MM-Bench (Liu et al., 2024d), MM-Vet (Yu et al., 2024), and five visual hallucanation benchmarks: CHAIR (Rohrbach et al., 2018), POPE (Li et al., 2023), AMBER (Wang et al., 2023), MMHal-Bench (Sun et al., 2023), HallBench (Guan et al., 2024). Besides, we also include four benchmarks in augmented visual domains: MMIU (Meng et al., 2024), MuirBench (Wang et al., 2024a), MVBench (Li et al., 2024), and Video-MME (Fu et al., 2025). For detailed experimental settings, configurations, and additional results, please refer to Appendix D.1.

### 4.1 MAIN RESULTS

**Performance on Comprehensive and Hallucination Benchmarks.** For comprehensive assessments of our proposed ViF, we first compare the results on six base VLMs, namely, LLaVA-v1.5-

Table 2: Results across eight comprehensive and hallucination benchmarks. * indicates implementation with Key-Norm, while others use attention scores. The best and the second best values with our method are **bolded** and underlined respectively, and the rightmost column shows the average results. For identical values, we compare the following digit after the decimal point.

| MAS Structure | Base Agent | MME↑ | MM Bench↑ | MM-Vet↑ | CHAIR↓ | POPE↑ | AMBER↑ | MMHal-Bench↑ | Hall Bench↑ | Avg.↑ |
|---|---|---|---|---|---|---|---|---|---|---|
| Linear | LLaVA-v1.5-7B | 1516.2 | 66.1 | 32.4 | 52.4 | 86.7 | 85.2 | 40.5 | 48.0 | |
| | **+Ours** | 1531.8 ↑15.6 | 67.8 ↑**1.7** | 34.6 ↑2.2 | 51.7 ↓0.7 | 88.6 ↑**1.9** | 87.8 ↑2.6 | 42.8 ↑2.3 | 50.6 ↑**2.6** | ↑ **3.5%** |
| | LLaVA-v1.6-7B | 1511.7 | 68.7 | 36.9 | 45.5 | 86.7 | 88.5 | 43.4 | 52.6 | |
| | **+Ours** | 1524.3 ↑12.6 | 69.4 ↑0.7 | 38.2 ↑1.3 | 43.6 ↓**1.9** | 88.2 ↑1.5 | 91.5 ↑**3.0** | 46.0 ↑2.6 | 54.7 ↑2.1 | ↑ **3.1%** |
| | LLaVA-NeXT-7B | 1567.0 | 72.1 | 47.1 | 44.6 | 88.6 | 87.0 | 45.9 | 52.9 | |
| | **+Ours** | 1585.2 ↑**18.2** | 73.5 ↑1.4 | 49.3 ↑2.2 | 42.9 ↓1.7 | 90.4 ↑1.8 | 89.3 ↑2.3 | 48.1 ↑2.2 | 55.3 ↑2.4 | ↑ **3.2%** |
| | LLaVA-OV-7B | 1587.6 | 82.3 | 58.4 | 38.3 | 91.4 | 91.3 | 47.8 | 53.7 | |
| | **+Ours*** | 1598.8 ↑11.2 | 83.4 ↑1.1 | 59.9 ↑1.5 | **37.2** ↓1.1 | **93.0** ↑1.6 | 93.9 ↑2.6 | 49.6 ↑1.8 | 56.1 ↑2.4 | ↑ **2.6%** |
| | Qwen2-VL-7B | 1686.4 | 81.9 | 63.3 | 38.4 | 90.5 | 91.2 | 48.2 | 51.9 | |
| | **+Ours*** | 1699.6 ↑13.2 | 82.4 ↑0.5 | 65.2 ↑1.9 | 37.8 ↓0.6 | 91.7 ↑1.2 | 94.0 ↑2.8 | 50.2 ↑2.0 | 54.4 ↑2.5 | ↑ **2.4%** |
| | Qwen2.5-VL-7B | 1730.4 | 83.9 | 67.3 | 38.6 | 89.9 | 92.8 | 51.6 | 53.8 | |
| | **+Ours*** | **1746.2** ↑15.8 | **85.3** ↑1.4 | **69.1** ↑1.8 | 37.6 ↓1.0 | 91.4 ↑1.5 | **94.4** ↑1.6 | **53.7** ↑2.1 | **56.2** ↑2.4 | ↑ **2.5%** |
| Layered | LLaVA-v1.5-7B | 1512.5 | 63.6 | 30.6 | 49.0 | 86.0 | 83.5 | 41.3 | 46.6 | |
| | **+Ours** | 1520.9 ↑8.4 | 64.7 ↑1.1 | 32.0 ↑1.4 | 49.3 ↑0.3 | 87.6 ↑1.6 | 86.1 ↑2.6 | 43.7 ↑2.4 | 48.9 ↑2.3 | ↑ **2.7%** |
| | LLaVA-v1.6-7B | 1508.0 | 66.6 | 35.1 | 44.2 | 86.8 | 85.2 | 42.0 | 48.2 | |
| | **+Ours** | 1518.9 ↑10.9 | 68.4 ↑1.8 | 37.5 ↑2.4 | 42.7 ↓1.5 | 87.9 ↑1.1 | 87.1 ↑1.9 | 43.6 ↑1.6 | 50.4 ↑2.2 | ↑ **3.2%** |
| | LLaVA-NeXT-7B | 1555.2 | 69.2 | 44.0 | 44.2 | 87.0 | 85.6 | 44.7 | 49.5 | |
| | **+Ours** | 1571.3 ↑**16.1** | 70.7 ↑1.5 | 46.5 ↑2.5 | 42.5 ↓1.7 | 89.2 ↑2.2 | 87.7 ↑2.1 | 47.2 ↑2.5 | 51.6 ↑2.1 | ↑ **3.5%** |
| | LLaVA-OV-7B | 1584.2 | 80.6 | 57.9 | 37.6 | 89.9 | 89.1 | 45.2 | 52.7 | |
| | **+Ours*** | 1596.7 ↑12.5 | 82.0 ↑1.4 | 59.5 ↑1.6 | 36.4 ↓1.2 | **91.5** ↑1.6 | 90.9 ↑1.8 | 46.9 ↑1.7 | **54.5** ↑1.8 | ↑ **2.5%** |
| | Qwen2-VL-7B | 1679.0 | 79.3 | 61.5 | 37.7 | 88.6 | 88.5 | 45.9 | 49.1 | |
| | **+Ours*** | 1692.6 ↑13.6 | 80.6 ↑1.3 | 63.4 ↑1.9 | 37.1 ↓0.6 | 90.5 ↑1.9 | 90.6 ↑2.1 | 48.1 ↑2.2 | 50.9 ↑1.8 | ↑ **2.5%** |
| | Qwen2.5-VL-7B | 1722.5 | 81.0 | 62.1 | 36.8 | 87.5 | 90.1 | 47.0 | 51.2 | |
| | **+Ours*** | **1737.0** ↑14.5 | **82.4** ↑1.4 | **63.8** ↑1.7 | **36.0** ↓0.8 | 89.6 ↑2.1 | **91.7** ↑1.6 | 49.3 ↑2.3 | 53.2 ↑2.0 | ↑ **2.6%** |
| Random | LLaVA-v1.5-7B | 1519.6 | 67.1 | 33.1 | 49.4 | 88.3 | 89.0 | 44.6 | 52.8 | |
| | **+Ours** | 1537.6 ↑18.0 | 68.4 ↑1.3 | 34.7 ↑**1.6** | 49.8 ↑0.4 | 90.2 ↑1.9 | 92.2 ↑3.2 | 47.0 ↑2.4 | 55.0 ↑2.2 | ↑ **2.8%** |
| | LLaVA-v1.6-7B | 1519.8 | 69.0 | 36.9 | 43.9 | 88.6 | 91.3 | 44.3 | 54.9 | |
| | **+Ours** | 1534.4 ↑14.6 | 69.7 ↑0.7 | 38.0 ↑1.1 | 41.8 ↓2.1 | 89.7 ↑1.1 | 94.0 ↑2.7 | 46.8 ↑2.5 | 57.3 ↑2.4 | ↑ **3.1%** |
| | LLaVA-NeXT-7B | 1576.2 | 73.0 | 49.2 | 43.4 | 90.4 | 89.2 | 47.2 | 55.4 | |
| | **+Ours** | 1596.1 ↑**19.9** | 75.3 ↑2.3 | 50.1 ↑0.9 | 41.6 ↓1.8 | 93.0 ↑2.6 | 92.9 ↑**3.7** | 49.3 ↑2.1 | 58.4 ↑**3.0** | ↑ **3.5%** |
| | LLaVA-OV-7B | 1590.1 | 83.7 | 58.5 | 37.7 | 92.5 | 91.9 | 46.3 | 56.2 | |
| | **+Ours*** | 1605.2 ↑15.1 | 85.1 ↑1.4 | 59.7 ↑1.2 | 37.1 ↓0.6 | **94.1** ↑1.6 | 94.9 ↑3.0 | 48.5 ↑2.2 | 59.1 ↑2.9 | ↑ **2.6%** |
| | Qwen2-VL-7B | 1690.1 | 84.1 | 64.4 | 38.1 | 90.8 | 92.3 | 46.5 | 53.7 | |
| | **+Ours*** | 1703.1 ↑13.0 | 84.9 ↑0.8 | 65.2 ↑0.8 | 37.2 ↓0.9 | 92.7 ↑1.9 | 95.4 ↑3.1 | 48.4 ↑1.9 | 56.3 ↑2.6 | ↑ **2.5%** |
| | Qwen2.5-VL-7B | 1737.7 | 86.5 | 67.0 | 37.8 | 90.4 | 93.6 | 48.9 | 56.8 | |
| | **+Ours*** | **1756.8** ↑**19.1** | **88.4** ↑1.9 | **68.1** ↑1.1 | **37.0** ↓0.8 | 92.8 ↑2.4 | **95.8** ↑2.2 | 50.6 ↑1.7 | **59.5** ↑2.7 | ↑ **2.5%** |
| Circular | LLaVA-v1.5-7B | 1520.9 | 67.9 | 33.5 | 52.7 | 88.9 | 88.7 | 42.4 | 51.7 | |
| | **+Ours** | 1539.1 ↑18.2 | 68.4 ↑0.5 | 36.0 ↑2.5 | 51.3 ↓1.4 | 90.4 ↑1.5 | 91.6 ↑2.9 | 44.7 ↑2.3 | 54.1 ↑2.4 | ↑ **3.4%** |
| | LLaVA-v1.6-7B | 1519.5 | 69.7 | 37.7 | 42.7 | 88.5 | 91.2 | 43.4 | 53.8 | |
| | **+Ours** | 1537.1 ↑17.6 | 71.3 ↑**1.6** | 39.3 ↑1.6 | 40.7 ↓2.0 | 90.1 ↑1.6 | 93.8 ↑2.6 | 46.0 ↑2.6 | 56.1 ↑2.3 | ↑ **3.5%** |
| | LLaVA-NeXT-7B | 1580.5 | 73.2 | 49.5 | 43.0 | 91.0 | 89.4 | 47.9 | 53.1 | |
| | **+Ours** | 1599.5 ↑**19.0** | 74.6 ↑1.4 | 51.8 ↑2.3 | 41.2 ↓1.8 | 93.3 ↑2.3 | 92.7 ↑3.3 | 51.1 ↑3.2 | 55.7 ↑2.6 | ↑ **3.8%** |
| | LLaVA-OV-7B | 1592.8 | 84.0 | 59.1 | 38.7 | 92.8 | 92.2 | 47.3 | 54.6 | |
| | **+Ours*** | 1606.1 ↑13.3 | 84.6 ↑0.6 | 60.2 ↑1.1 | **36.9** ↓1.8 | **94.0** ↑1.2 | 95.0 ↑2.8 | 49.4 ↑2.1 | 56.9 ↑2.3 | ↑ **2.7%** |
| | Qwen2-VL-7B | 1692.8 | 83.3 | 63.9 | 38.1 | 91.6 | 92.8 | 47.7 | 52.4 | |
| | **+Ours*** | 1706.3 ↑13.5 | 84.1 ↑0.8 | 65.1 ↑1.2 | 37.2 ↓0.9 | 93.3 ↑1.7 | 94.8 ↑2.0 | 50.2 ↑2.5 | 54.6 ↑2.2 | ↑ **2.4%** |
| | Qwen2.5-VL-7B | 1738.1 | 85.2 | 66.7 | 38.2 | 91.3 | 93.5 | 50.1 | 54.9 | |
| | **+Ours*** | **1756.2** ↑18.1 | **86.6** ↑1.4 | **68.1** ↑1.4 | 37.4 ↓0.8 | 93.4 ↑2.1 | **95.9** ↑2.4 | **52.5** ↑2.4 | **57.3** ↑2.4 | ↑ **2.6%** |

Table 3: Results of larger-size models on circular MAS structure.

| Base Agent | MME↑ | MM Bench↑ | MM-Vet↑ | CHAIR↓ | POPE↑ | AMBER↑ | MMHal-Bench↑ | Hall Bench↑ | Avg.↑ |
|---|---|---|---|---|---|---|---|---|---|
| LLaVA-1.5-13B | 1528.7 | 70.2 | 38.3 | 40.8 | 90.0 | 89.6 | 44.7 | 52.9 | |
| **+Ours** | 1547.6 ↑18.9 | 71.1 ↑0.9 | 40.5 ↑2.2 | 39.1 ↓1.7 | 92.4 ↑2.4 | 92.7 ↑**3.1** | 47.2 ↑2.5 | 55.3 ↑2.4 | ↑ **3.6%** |
| LLaVA-NeXT-13B | 1583.5 | 68.8 | 42.3 | 36.0 | 91.9 | 92.4 | 48.2 | 54.3 | |
| **+Ours** | 1602.6 ↑19.1 | 70.1 ↑1.3 | 44.5 ↑2.2 | 34.2 ↓1.8 | 93.7 ↑1.8 | 95.4 ↑3.0 | 50.8 ↑2.6 | 56.8 ↑2.5 | ↑ **3.6%** |
| LLaVA-NeXT-34B | 1644.9 | 78.6 | 54.6 | 27.6 | 91.4 | 94.1 | 48.9 | 55.0 | |
| **+Ours** | 1670.8 ↑**25.9** | 80.9 ↑**2.3** | 57.0 ↑2.4 | 25.4 ↓2.2 | 93.6 ↑2.2 | 96.3 ↑2.2 | 52.4 ↑**3.5** | 57.8 ↑**2.8** | ↑ **4.4%** |
| Qwen2.5-VL-32B | 1886.1 | 87.4 | 69.8 | 24.4 | 92.5 | 94.0 | 52.1 | 56.9 | |
| **+Ours*** | **1906.2** ↑20.1 | **89.2** ↑1.8 | **71.9** ↑2.1 | **22.3** ↓2.1 | **94.0** ↑1.5 | **96.7** ↑2.7 | **55.1** ↑3.0 | **60.1** ↑3.2 | ↑ **4.1%** |

7B (Liu et al., 2024a), LLaVA-v1.6-7B, LLaVA-NeXT-7B (Liu et al., 2024b), LLaVA-OV-7B (Li et al., 2025a), Qwen2-VL-7B (Wang et al., 2024b), and Qwen2.5-VL-7B (Bai et al., 2025). We choose four common MAS structures, including linear (Hong et al., 2024), layered (Ishibashi & Nishimura, 2024), random (Qian et al., 2025), and circular (Qian et al., 2025) structures. As

Table 4: Results across four augmented visual benchmarks on circular MAS structure, including multi-image and video based scenarios.

| Base Agent | MMIU↑ | MuirBench↑ | MVBench↑ | Video-MME↑ | Avg.↑ |
|---|---|---|---|---|---|
| LLaVA-NeXT-7B | 31.6 | 42.6 | 49.2 | 60.4 | |
| +Ours | 33.9 ↑2.3 | 44.3 ↑1.7 | 52.0 ↑2.8 | 61.9 ↑1.5 | ↑4.9 |
| LLaVA-OV-7B | 36.9 | 54.2 | 56.1 | 67.8 | |
| +Ours* | 39.6 ↑2.7 | 55.5 ↑1.3 | 58.3 ↑2.2 | 68.8 ↑1.0 | ↑3.8 |
| Qwen2-VL-7B | 45.5 | 62.8 | 69.8 | 70.6 | |
| +Ours* | 47.7 ↑2.2 | 64.0 ↑1.2 | 71.0 ↑1.2 | 71.7 ↑1.1 | ↑2.5 |
| Qwen2.5-VL-7B | 47.4 | 64.0 | 72.3 | 73.4 | |
| +Ours* | 49.3 ↑1.9 | 65.0 ↑1.0 | 73.6 ↑1.3 | 74.0 ↑0.6 | ↑2.0 |

Table 5: Evaluations of multi-agent hallucination snowballing with proposed *HS* metric.

| MAS Structure | CHAIR↓ | POPE↓ | AMBER↓ | MMHal-Bench↓ | Hall Bench↓ | Avg.↓ |
|---|---|---|---|---|---|---|
| Linear | 17.2 | 25.7 | 26.8 | 35.3 | 40.2 | |
| | 12.4 ↓4.8 | 16.4 ↓9.3 | 15.9 ↓10.9 | 22.6 ↓12.7 | 24.8 ↓15.4 | ↓ **35.8%** |
| Layered | 12.7 | 21.5 | 20.5 | 31.6 | 36.4 | |
| | 10.6 ↓2.1 | 13.9 ↓7.6 | 13.1 ↓7.4 | 19.5 ↓12.1 | 21.3 ↓15.1 | ↓ **33.6%** |
| Circular | 18.9 | 29.1 | 31.1 | 40.8 | 47.4 | |
| | 12.8 ↓6.1 | 17.0 ↓12.1 | 17.7 ↓13.4 | 24.1 ↓16.7 | 27.8 ↓19.6 | ↓ **39.8%** |
| Random | 15.5 | 23.4 | 23.8 | 36.8 | 42.5 | |
| | 10.3 ↓5.2 | 15.0 ↓8.4 | 16.4 ↓7.4 | 21.2 ↓15.6 | 25.6 ↓16.9 | ↓ **36.5%** |

Table 6: Comparison results of other SOTA methods and ours on LLaVA-NeXT-7B and circular MAS structure. *Orig.* represents the original evaluation metric, and *HS* is our proposed one.

| | CHAIR | | POPE | | AMBER | | MMHal-Bench | | HallBench | | Avg. | |
|---|---|---|---|---|---|---|---|---|---|---|---|---|
| | *Orig.↓* | *HS↓* | *Orig.↑* | *HS↓* | *Orig.↑* | *HS↓* | *Orig.↑* | *HS↓* | *Orig.↑* | *HS↓* | *Orig.↑* | *HS↓* |
| Baseline | 43.0 | 18.9 | 91.0 | 29.1 | 89.4 | 31.1 | 47.9 | 40.8 | 53.1 | 47.4 | | |
| MemVR | 43.8 ↑0.8 | 20.6 ↑1.7 | 90.5 ↓0.5 | 31.2 ↑2.1 | 88.9 ↓0.5 | 34.4 ↑3.3 | 44.8 ↓3.1 | 58.6 ↑17.8 | 49.2 ↓3.9 | 57.6 ↑10.2 | ↓ **2.6%** | ↑ **18.4%** |
| VISTA | 43.4 ↑0.4 | 19.0 ↑0.1 | 91.2 ↑0.2 | 27.8 ↓1.3 | 90.5 ↑1.1 | 28.3 ↓2.8 | 46.3 ↓1.6 | 47.4 ↑6.6 | 50.7 ↓2.4 | 53.3 ↑5.9 | ↓ **1.1%** | ↑ **3.1%** |
| FarSight | 42.1 ↓0.9 | 17.7 ↓1.2 | 91.9 ↑0.9 | 22.7 ↓6.4 | 91.0 ↑1.6 | 26.6 ↓4.5 | 47.4 ↓0.5 | 42.9 ↑2.1 | 51.9 ↓1.2 | 52.4 ↑5.0 | ↓ **0.5%** | ↑ **5.4%** |
| DeCo | 42.6 ↓0.4 | 18.2 ↓0.7 | 91.3 ↑0.3 | 25.1 ↓4.0 | 91.6 ↑2.2 | 24.3 ↓6.8 | 47.0 ↓0.9 | 44.1 ↑3.3 | 50.4 ↓2.7 | 53.0 ↑5.6 | ↓ **1.0%** | ↓ **3.8%** |
| TAME | 42.1 ↓0.9 | 18.8 ↓0.1 | 91.4 ↑0.4 | 22.8 ↓6.3 | 91.9 ↑2.5 | 22.7 ↓8.4 | 46.5 ↓1.4 | 47.8 ↑7.0 | 49.9 ↓3.2 | 53.8 ↑6.4 | ↓ **1.6%** | ↓ **3.7%** |
| **Ours** | 41.2 ↓1.8 | 12.8 ↓6.1 | 93.3 ↑2.3 | 17.0 ↓12.1 | 92.7 ↑3.3 | 17.7 ↓13.4 | 51.1 ↑3.2 | 24.1 ↓16.7 | 55.7 ↑2.6 | 27.8 ↓19.6 | ↑ **3.8%** | ↓ **39.8%** |

mentioned in Appendix C.1, we primarily follow the multi-agent collaboration strategy with linear increased context length, allowing the scaling of MAS. As demonstrated in Table 2, our ViF consistently enhances the average performance of the six baselines by 2.4-3.8%, verifying the compatibility of our method on various MAS structures based on arbitrary base VLMs. Notably, on the MMHal-Bench and HallBench benchmarks, which are more sophisticated and have unsatisfactory baseline performance, our ViF achieves over 4% average improvement. When applied to the circular structure, which is hallucination-concentrated with densest collaborations and interactions among agents, our ViF dramatically reduces hallucination snowballing and further improves the performance by 3% among the six base models, compared to the other three selected structures.

As reported in Table 3, we also analyze the performance of our proposed ViF on the scaled-up models with higher parameters. It is observed that when equipped with our ViF, the larger base models featuring more than 30B parameters, *e.g.*, LLaVA-NeXT-34B and Qwen-2.5-VL-32B, exhibit greater enhancement than all smaller ones, improving by more than 4% across all benchmarks. This indicates that our model-agnostic method effectively improves their comprehensive performance, likely because larger-parameter baselines possess stronger fundamental capabilities, and our approach specifically unlocks their latent potential in multi-agent scenarios.

**Performance on Augmented Visual Benchmarks.** We include additional four benchmarks of two augmented visual domains, including two multi-image based benchmarks: MMIU, MuirBench; video based two benchmarks: MVBench, Video-MME. As presented in Table 4, our ViF method exhibits significant improvements relative to the base models across multi-image and video scenarios. Specifically, it yields an average 2.0–4.9% performance improvement across the four base models and four additional benchmarks, demonstrating robust performance in multiple visual scenarios.

**Multi-Agent Hallucination Snowballing Mitigation.** In addition to the results of original metrics, we attempt to assess the level of hallucination snowballing in MAS quantitatively. Thus, we formally define a hallucination snowballing score (*HS*) as in Equation 7, measuring both the hallucination level and propagation in MAS. As reported in Table 5, adding our ViF reduces at least 30% *HS* score on the average of five hallucination benchmarks, significantly mitigating the hallucination propagation from the textual flow of visual contents. Notably, the layered structure suffers the least from the detrimental snowballing, while in circular structure, where the initial hallucination snowballing is the most serious, the reduction of the score from our method is almost 40%.

**Comparison Results.** We compare the results of another five model-agnostic and token-wise hallucination mitigation methodologies in multi-agent contexts, *i.e.*, MemVR (Zou et al., 2025),

VISTA (Li et al., 2025c), FarSight (Tang et al., 2025b), DeCo (Wang et al., 2025), and TAME (Tang et al., 2025a). Specifically, we retain the multi-agent experimental settings unchanged and apply these methodologies to the base model. These counterparts restrict the deepening of intrinsic hallucinations in a single model to some extent, however, in multi-agent scenarios, the propagation of visual contents via textual flow still introduces a vision-to-text cognitive bias and fails to restrain the snowballing of multi-agent hallucinations commendably.

As shown in Table 6, our introduced ViF approach achieves distinctly superior performance among all the benchmarks on both their original metrics and our proposed *HS* score. It obtains at least 4.2% enhancements in original metrics and 34.4% in *HS* score on average, compared to other methods tailored for single model hallucination mitigation. Although these counterparts are impressively efficacious in single VLM, their performance is compromised when applied to MAS, because of the failure to deal with hallucination propagation among agents and further snowballing. Surprisingly, in MAS environments, the results of these counterparts are even inferior to the baseline, especially on challenging ones. This counterintuitive observation is likely because they modify the initial paradigm of decoding or attention in VLMs, but retain the textual flow to relay visual information, which amplifies the preference for text over vision tokens. Our method, adopting visual flow to relay information among agents, cuts the *HS* score almost in half and delivers tangible improvements in the mitigation of hallucination snowballing.

## 4.2 ADDITIONAL ANALYSES

**Impact of the Number of Agent Turns.** To achieve satisfactory completion, typically, MAS necessitates a greater number of agent turns in more complicated and challenging tasks. However, the hallucination snowballing effect restricts the multi-turn collaboration among agents, where hallucinations are amplified and propagated, leading to suboptimal performance. Therefore, we compare our ViF with baselines and other counterparts to assess the impact of the number of agent turns.

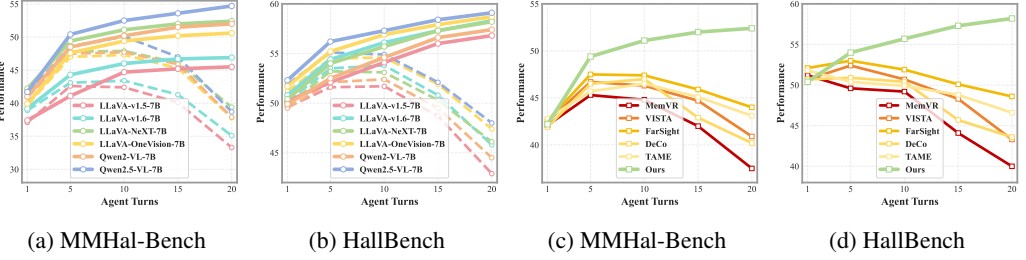

|       (a) MMHal-Bench       |       (b) HallBench       |       (c) MMHal-Bench       |       (d) HallBench       |

Figure 7: Impact of the number of agent turns. In (a) and (b), straight and dashed lines are the results with or without our ViF on various baselines and circular MAS structure, respectively. (c) and (d) show the results between other counterparts and our method based on LLaVA-NeXT-7B.

As demonstrated in Figure 7, our method maintains an upward trend in performance as the number of agent turns increases, while both other contrast methods and baselines experience performance degradation instead. More precisely, when the agent turn is set to one, which is equivalent to a single-agent context, ViF exhibits only a marginal improvement over the baselines, and falls behind some other methods designed for hallucination mitigation in a single model. As the turning trends of the baselines in Figure 7a and 7b show, the performance begins to deteriorate when the turns are only increased to 5, and at the 20th turn, their performance is even further less than that of a single agent. Further compare with other methods as illustrated in Figure 7c and 7d, although hallucinations are mitigated in early turns to some extent, the hallucination snowballing phenomenon still suffers in later turns, essentially limiting the multi-agent collaboration and inhibiting the potential of MAS.

**Ablation and Sensitivity Analyses.** To verify the effectiveness of each component in our ViF, we perform ablations on the visual relay tokens and the attention reallocation. As reported in Table 7, the improvement from visual flow to relay information is prominent, and the results are still better than most comparison methods even when ablating half of the visual relay tokens, showcasing excellent robustness. The reallocation mechanism further optimizes the attention distribution among different tokens and activates visual relay tokens, which is beneficial to our designs of visual relay flow.

Table 8: Efficiency comparison between our ViF and the base models on the circular MAS architecture. All base models are evaluated in the multi-agent environment to quantify the additional latency introduced by ViF, including the average *latency* (seconds), and the average floating point operations, *i.e.*, *FLOPs* (T).

| Base Agent | CHAIR | | POPE | | AMBER | | MMHal-Bench | | HallBench | |
|---|---|---|---|---|---|---|---|---|---|---|
| | *Latency↓* | *FLOPs↓* | *Latency↓* | *FLOPs↓* | *Latency↓* | *FLOPs↓* | *Latency↓* | *FLOPs↓* | *Latency↓* | *FLOPs↓* |
| LLaVA-NeXT-7B | 3.16 | 157.3 | 2.46 | 103.4 | 2.79 | 127.2 | 3.48 | 184.0 | 3.91 | 248.3 |
| **+Ours** | 3.47 | 168.5 | 2.79 | 115.7 | 3.10 | 138.9 | 3.83 | 197.6 | 4.23 | 260.3 |
| LLaVA-NeXT-13B | 5.88 | 308.5 | 5.63 | 279.0 | 5.93 | 310.0 | 6.33 | 357.4 | 7.64 | 386.8 |
| **+Ours** | 6.17 | 320.6 | 5.91 | 289.6 | 6.25 | 321.2 | 6.67 | 372.8 | 8.03 | 399.6 |
| LLaVA-NeXT-34B | 8.80 | 417.2 | 8.49 | 387.3 | 8.61 | 408.7 | 9.41 | 444.1 | 11.06 | 478.1 |
| **+Ours** | 9.09 | 419.1 | 8.79 | 398.8 | 8.92 | 419.8 | 9.75 | 457.7 | 11.41 | 493.8 |

Furthermore, as listed in Table 12, 13, and 14, we conduct an analysis of the sensitivity of key hyper-parameters, *i.e.*, the salience threshold $\omega$, the temperature scaling $\tau$, and the reallocation coefficient $\alpha$, quantitatively assess their impact on model performance, and determine a rational set of values.

**Efficiency.** To assess the inference efficiency, particularly in multi-agent contexts, we first compare the time and computational overheads of our proposed ViF with those of the base mod-

Table 7: Ablation study on LLaVA-NeXT-7B and circular MAS structure, verifying the effectiveness of visual relay tokens and attention reallocation.

| Setting | CHAIR↓ | POPE↑ | AMBER↑ | MMHal-Bench↓ | Hall Bench↓ |
|---|---|---|---|---|---|
| *w/o* Relay Token (25%) | 41.4 (+0.2) | 92.9 (-0.4) | 92.3 (-0.4) | 50.7 (-0.4) | 55.2 (-0.5) |
| *w/o* Relay Token (50%) | 42.3 (+1.1) | 92.0 (-1.3) | 91.6 (-1.1) | 49.8 (-1.3) | 54.8 (-0.9) |
| *w/o* Relay Token (75%) | 42.6 (+1.4) | 91.7 (-1.6) | 91.1 (-1.6) | 49.1 (-2.0) | 54.1 (-1.6) |
| *w/o* Reallocation (Middle) | 41.7 (+0.5) | 92.1 (-1.2) | 91.4 (-1.3) | 49.9 (-1.2) | 54.4 (-1.3) |
| *w/o* Reallocation (Deep) | 41.4 (+0.2) | 92.7 (-0.6) | 92.5 (-0.2) | 50.9 (-0.2) | 55.0 (-0.7) |
| *w/o* Reallocation | 42.2 (+1.0) | 91.9 (-1.4) | 91.5 (-1.2) | 49.6 (-1.5) | 54.2 (-1.5) |
| **Ours** | **41.2** | **93.3** | **92.7** | **51.1** | **55.7** |

els. As reported in Table 8, our ViF exhibits high efficiency with moderate overheads, incurring an additional 8.1-13.4% inference latency and 4.8-11.9% computational costs (measured by *FLOPs*) over the base model across five selected benchmarks. These extra overheads mainly stem from the intrinsic components of ViF. Thus, the extra overhead remains stable across base models of varying scales and exhibits only slightly linear increase. Notably, the additional latency and computation are even more negligible for larger models, which are less than 4% and 3% on LLaVA-NeXT-34B. Furthermore, as presented in Table 15, the time and computational overhead of our ViF remain efficient when feeding visual images with varying resolutions and across different agent turns.

## 5 CONCLUSION

We unveil the phenomenon of multi-agent visual hallucination snowballing existing in MAS, where subsequent agents progressively amplify errors originating in a single agent through textual information flow that relays visual messages. Based on extensive analyses, the essence of hallucination snowballing lies in a subset of vision tokens with an unimodal attention peak, well-preserving the visual information, but these tokens gradually diminish with the increase in the agent turns. To alleviate this problem, a model-agnostic method named ViF is proposed, which redefines the visual information flow in MAS. Specifically, we introduce a visual flow to relay visual messages based on the selected unimodal vision tokens and utilize attention reallocation to optimize this pattern. Comprehensive experiments indicate that this novel paradigm is effective, robust, and compatible, paving the way for more efficient inter-agent visual information relay and more sophisticated MAS.

**Limitations.** Although we conduct experiments on a total of ten models with different sizes, which verifies the robustness of compatibility of our proposed method, more experiments are still recommended. For example, the results on smaller size VLMs, *e.g.*, 3B, and also larger baselines, *e.g.*, 72B, could provide further evidence. Besides, the inclusion of more series of baselines, such as InternVL series (Chen et al., 2024b;a; Zhu et al., 2025), Llama 3 (Grattafiori et al., 2024), Ovis series (Lu et al., 2024; 2025), and MiniMCP (Hu et al., 2024), is also beneficial. Furthermore, more effective and complementary combinations of our ViF with other hallucination mitigation strategies, as well as more optimal vision token selection, warrant further exploration.

## 6 ACKNOWLEDGEMENT

This work was supported by the National Natural Science Foundation of China under Grant No. 62320106007, and by NUS Start-up Grant A-0010106-00-00.

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

## APPENDIX

## A  RELATED WORKS

**Visual Hallucination.** The tendency of VLMs to generate plausible but non-factual or unsupported content, *i.e.*, visual hallucination, is well-documented in previous works. A common remedy has

been to retrain or fine-tune models to better align outputs with ground truth (Zhou et al., 2024; Zhai et al., 2023; Yue et al., 2024), but these solutions often demand extensive training resources and additional data. Consequently, interest has grown in training-free techniques, including self-feedback correction (Lee et al., 2024; Yin et al., 2023), leveraging auxiliary models for external knowledge integration (Yang et al., 2024), and modifying decoding procedures (Wang et al., 2024c; Zou et al., 2025; Wang et al., 2025; Tang et al., 2025b; Li et al., 2025c; Tang et al., 2025a; Yin et al., 2025; Liu et al., 2023). In contrast to these papers that focus on a single VLM agent, it is not sufficient to address the failure mode of hallucination snowballing that emerges in multi-agent collaboration, which is the core focus of our paper.

**Attention in VLM-Based Agents.** The hallucination problem of VLMs can be mainly attributed to and indicated by the attention mechanism. Earlier work found that LVLMs tend to attend to broad, global image cues and miss prompt-relevant details (Darcet et al., 2024; Gong et al., 2024; An et al., 2024), a behavior often traced to the Vision Transformer encoder (Alexey, 2020). To address this, some methods boost attention weights for pertinent image tokens (Liu et al., 2024c), others select or filter informative visual features and apply contrastive decoding to suppress hallucinations (Huo et al., 2025). Our work presents an extensive study on the attention allocation token and layer-wise analysis to provide a better understanding of how multi-agent pipelines lose visual fidelity during inference turns. Based on this, we further introduce a novel visual flow to alleviate hallucination snowballing in multi-agent systems.

# B  REQUISITE ANALYSES

## B.1  SETTINGS

**Attention Allocation.** In Section 2, we first calculate the attention allocations of four tokens in different layers among different agent turns. Formally, we first denote the whole token set as $\mathcal{T}$, consisting of vision token subset $\mathcal{V}$, instruction token subset $\mathcal{I}$, system token subset $\mathcal{S}$, and output token subset $O$. The attention matrix is obtained as Equation 2, and each attention score $s_{i,j}$ indicates the attention from the $i$th token to the $j$th token. Thus, the attention allocation of a specific token type should be the sum of the attention score where the target is this token, which could be calculated as follows:

$$Allocation_{token\_type} = \sum_{i \in \mathcal{T}} \sum_{j \in \mathcal{T}} \mathcal{A}_l\left(i, j\right) \circ M_{token\_type} = \sum_{i \in \mathcal{T}} \sum_{j \in \mathcal{T}} s_{i,j}, \tag{5}$$

$$M_{token\_type} = \mathbb{I}\left(i \in \mathcal{T}, j \in \mathcal{T}_{token\_type}\right) \tag{6}$$

where $\mathcal{A}$ denotes the average attention matrix in all attention heads of the $l$th layer in this context. The attention allocation of specific tokens explicitly represents the focus in each layer of the model when understanding the task and outputting the responses.

**Dropping Tokens in Certain Layers.** To drop subsets of vision tokens in shallow, middle, or deep layers, we set the hidden states of the subset in specific layers to zero instead of physical removal, because the latter changes sequence length and disrupts sequence alignment of the attention mechanism. Moreover, the implementation is mainly modified from (Liu et al., 2024b).

**Token Selection.** As described in Section 2, we select five subsets of vision tokens, and here we elaborate on the selection rules for each subset: (1) Random Tokens: we randomly select from all vision tokens, and limit the number of tokens in the subset to the average number in the other four subsets. Besides, we re-select the random tokens if the size of the largest connected component of selected tokens exceeds 10% of the total selected tokens, to avoid selecting centralized tokens that destroy randomness; (2) Inactive Tokens: we first calculate the average attention value across all layers for each token, then select tokens whose attention values are below the lower quartile and whose fluctuation does not exceed 20%; (3) Rise Tokens and (4) Fall Tokens: we select the tokens with gradually upward or downward attention allocation in the consecutive layers. To filter out insignificant fluctuations and better reflect the overall attention trend of each token, we utilize a tolerance threshold. When deviations from the trend in the opposite direction do not exceed this threshold, we still consider it as maintaining the original trend; (5) Unimodal Tokens: we select tokens with attention allocation of a unimodal distribution, whose peak surpasses the salience threshold $\omega$.

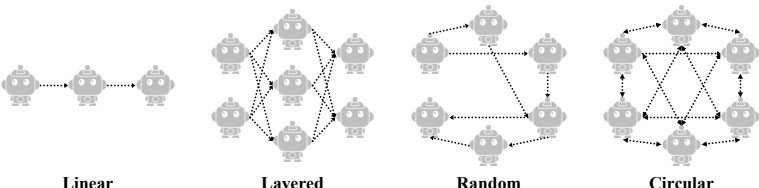

Figure 8: The four structures of MAS in our experiments.

## B.2 ADDITIONAL RESULTS

As discussed in Section 2, we use the results of the LLaVA-NeXT-7B (Liu et al., 2024b) model on POPE (Li et al., 2023) as an example. Here, we provide results of different base VLMs to verify the generality of our insights and to avoid model-specific conclusions. The token-wise attention allocations of vision, instruction, system, and output tokens of six common VLMs are demonstrated in Figure 13; the results of dropping different subsets of vision tokens in the shallow, middle and deep layers are listed in Table 16; and the proportion of different vision tokens among different agent turns are demonstrated in Figure 14. These experimental results from the six models are consistent with the previous conclusions, demonstrating excellent generalization across various models.

## C METHODOLOGIES

### C.1 MAS STRUCTURES

Since our proposed ViF primarily centers on the snowballing of visual hallucinations in multi-agent contexts, we first briefly delineate the MAS structures defined herein. Existing MAS can be primarily categorized into two basic architectures (Guo et al., 2024): the centralized and the decentralized. The former are function-specialized, involving intricate collaborative workflows or functional division mechanisms. To mitigate such uncertainty, we adopt the latter decentralized ones, specifically incorporating four distributed structures that feature no central agent and a relatively straightforward structure; all agents are equal in status, save for sequential dependencies within the system. As illustrated in Figure 8, we include four particular sub-structures: linear structure (Hong et al., 2024), which implements a linear configuration for agent-mediated interactions; layered structure (Ishibashi & Nishimura, 2024), which comprises multiple hierarchical layers, where agent nodes within the current layer establish connections exclusively with those in the subsequent layer; random structure (Qian et al., 2025), which establishes random connections among agent nodes, where each agent may dynamically decide to redirect to the subsequent node based on current contextual information. Notably, this structure features unidirectional paths, thereby failing to ensure that a single node can reach all other nodes within the network; and circular structure (Qian et al., 2025), which utilizes fully-connected mesh, ensuring that each agent node can reach any other node in the system via at least one path. Intuitively, among the four MAS structures, circular ones have the densest collaborations and interactions among agents, and theoretically, the multi-agent hallucination snowballing effects are the most serious. Thus, all the experiments except the main results are conducted on the circular structure with more obvious visual hallucination snowballing.

Primarily drawing on the multi-agent collaboration framework (Qian et al., 2025), we adopt an interactive collaboration strategy in our MAS, one anchored in topological ordering of directed acyclic graphs. This strategy employs a dual-agent multi-round interaction model: within each edge of the network, in the iterative interaction, adjacent actors are assigned to nodes, and critics are assigned to edges. Specifically, the preceding actor first requests feedback; the critic then provides reflective suggestions and requests further refinement; and finally, the subsequent actor generates an optimized artifact. Through this process, the prior artifact is iteratively refined. Specifically, this collaboration expand the conventional single agent method to multi-agent environments and reduces the context length from quadratic growth to linear growth, allowing the collaborative scaling law in MAS. Consequently, this architectural design is adopted as the core MAS framework in this work.

Table 9: Results of the attention score based and other alternative strategies on LLaVA-NeXT-7B and circular MAS structure.

| Selection Strategy | MME↑ | MM Bench↑ | MM-Vet↑ | CHAIR↓ | POPE↑ | AMBER↑ | MMHal-Bench↑ | Hall Bench↑ |
|---|---|---|---|---|---|---|---|---|
| Value-Norm | 1585.6 (-13.9) | 72.3 (-2.3) | 49.7 (-2.1) | 43.4 (+2.2) | 90.5 (-2.8) | 90.1 (-2.6) | 49.0 (-2.1) | 53.6 (-2.1) |
| Value-Norm (+1 Buffer) | 1587.4 (-12.1) | 72.6 (-2.0) | 49.9 (-1.9) | 43.6 (+2.4) | 90.6 (-2.7) | 90.4 (-2.3) | 49.4 (-1.7) | 53.9 (-1.8) |
| Value-Norm (+3 Buffer) | 1588.9 (-10.6) | 72.7 (-1.9) | 50.3 (-1.5) | 43.1 (+1.9) | 90.8 (-2.5) | 90.3 (-2.4) | 49.2 (-1.9) | 53.9 (-1.8) |
| Value-Norm (+5 Buffer) | 1590.2 (-9.3) | 73.4 (-1.2) | 50.5 (-1.3) | 43.0 (+1.8) | 91.1 (-2.2) | 90.6 (-2.1) | 49.4 (-1.7) | 54.1 (-1.6) |
| Value-Norm (+8 Buffer) | 1589.8 (-9.7) | 73.3 (-1.3) | 50.7 (-1.1) | 43.3 (+2.1) | 90.9 (-2.4) | 90.8 (-1.9) | 49.6 (-1.5) | 54.4 (-1.3) |
| Key-Norm | 1593.0 (-6.5) | 74.0 (-0.6) | 51.6 (-0.2) | 41.5 (+0.3) | 92.9 (-0.4) | 92.1 (-0.6) | 50.6 (-0.5) | 55.1 (-0.6) |
| Key-Norm (+1 Buffer) | 1594.4 (-5.1) | 74.2 (-0.4) | 51.7 (-0.1) | 41.3 (+0.1) | 93.1 (-0.2) | 92.4 (-0.3) | 50.9 (-0.2) | 55.4 (-0.3) |
| Key-Norm (+3 Buffer) | 1595.9 (-3.6) | 74.3 (-0.3) | **51.9** (+0.1) | **41.2** (-0.0) | 93.1 (-0.2) | 92.6 (-0.1) | **51.2** (+0.1) | 55.5 (-0.2) |
| Key-Norm (+5 Buffer) | 1595.2 (-4.3) | 74.3 (-0.3) | 51.7 (-0.1) | 41.4 (+0.2) | 92.8 (-0.5) | 92.3 (-0.4) | 50.9 (-0.2) | 55.6 (-0.1) |
| Key-Norm (+8 Buffer) | 1594.7 (-4.8) | 74.4 (-0.2) | 51.5 (-0.3) | 41.6 (+0.4) | 92.9 (-0.4) | 92.3 (-0.4) | 50.8 (-0.3) | 55.4 (-0.3) |
| Attention Score | **1599.5** | **74.6** | 51.8 | 41.2 | **93.3** | **92.7** | 51.1 | **55.7** |

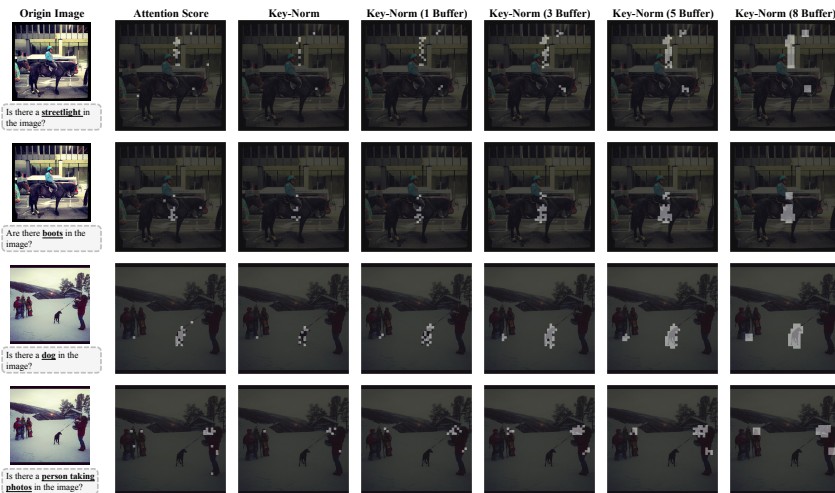

Figure 9: Comparisons of selected tokens using the attention scores and other alternative strategies.

## C.2 ALTERNATIVE OF ATTENTION SCORE BASED STRATEGY

Given that Flash-Attention 2/3 (Dao, 2024) are widely used in the latest VLMs, resulting in attention scores that are not explicitly stored and are not accessible, we design an alternative token selection strategy inspired by (Wen et al., 2025). Specifically, we utilize the $L_2$ norm of the key to replace the attention score, which reflects the feature strength of the token; a higher value of the norm indicates that the token is relatively more prominent and has more significant semantics. Unlike the strategy introduced in (Wen et al., 2025), which adopts $L_1$ norm, we choose $L_2$ norm to amplify the difference between tokens and promote the token selection. Statistically, the overlap of the initially selected tokens of the two strategies is more than 70%; however, the total amount of the Key-Norm based strategy is less than that of the other. Thus, we add buffer tokens of initially selected tokens, which surround the initially selected token of the $3 \times 3$ space.

To verify the effectiveness of the Key-Norm, we compare it with the attention score based strategy as well as Value-Norm, which is similar to our alternative. As illustrated in Table 9, we observe that the Key-Norm based strategies are superior to the Value-Norm based ones. Besides, selecting by Key-Norm with three buffer tokens almost achieves the same performance as attention scores, even surpassing them in partial benchmarks. For more intuitive results, we provide the visualization comparisons of selected tokens based on these strategies from real cases. As visualized in Figure 9, the initial Key-Norm based selection covers the most visual relay tokens compared to the attention score based selection; however, the former one is relatively more sparse, losing partially important visual semantics. Adding buffer tokens is a good solution, which selects the surrounding tokens and supplements the information of the visual flow. It is worth noting that we add three buffer tokens when using the alternative strategy in our experiments, thereby balancing accuracy and efficiency.

# D EXPERIMENTS

## D.1 SETTINGS

**Baselines.** To verify the generality, we totally adopt ten models covering from 7B to 34B in our experiments, which are listed as follows:

- LLaVA-v1.5 (Liu et al., 2024a) uses a two-layer MLP to connect image features into the word embedding space, and we choose the 7B and 13B models, which have 32 and 40 layers in the decoder of the LLM, respectively.
- LLaVA-v1.6 (Liu et al., 2024b) increases resolution limits for input images with augmented data. It is the early version of LLaVA-NeXT, which has 32 layers in the LLM.
- LLaVA-NeXT (Liu et al., 2024b) has significant improvements in reasoning, OCR, and world knowledge, remaining the same structure as LLaVA-v1.6. We include 7B, 13B, and 34B models in our experiments, which have 32, 40, and 60 layers in the decoder.
- LLaVA-OV (Li et al., 2025a) is the most powerful base VLM in the LLaVA family, supporting single-image-, multi-image, and video scenes simultaneously. It uses the Qwen2 (Team, 2024) as the LLM, which has 28 layers.
- Qwen2-VL (Wang et al., 2024b) introduces naive dynamic resolution and multi-modal rotary position embedding (M-RoPE), achieving impressive image and video understanding. Its 7B model has 28 decoder layers.
- Qwen2.5-VL (Bai et al., 2025) compresses the vision tokens with an MLP-based fuser, aligns M-RoPE and absolute time, and meticulously designs a three-stage training pipeline. We select the 7B and 32B models with 32 layers of the LLM.

**Benchmarks.** We evaluate our method on eight widely adopted benchmarks, including three comprehensive benchmarks and five hallucination benchmarks, which are presented as follows:

- MME (Yin et al., 2024) is the first comprehensive benchmark and measures the perception and cognitive abilities of 14 challenging subtasks. Moreover, the metric is the total score across all the subtasks.
- MMBench (Liu et al., 2024d) consists of multiple-choice questions to assess over twenty different ability dimensions, offering a hierarchical framework with three levels. During the assessment, GPT-4 serves as the final judge. In our experiments, we only include the English subset for evaluation.
- MM-Vet (Yu et al., 2024) is a comprehensive benchmark, which defines six core capacities and assesses them on complicated visual tasks. It provides a GPT-4 based evaluator for open-ended outputs.
- CHAIR (Rohrbach et al., 2018) is a captioning hallucination assessment benchmark, comparing the objects mentioned in the title with the objects actually existing in the image. Here, we utilize the $CHAIR_S$ metric, calculating the proportion of titles that contain at least one hallucinatory object.
- POPE (Li et al., 2023) merges several classic visual datasets, and generates binary questions of the existence of objects. Each image is paired with six questions, and we use the accuracy metric.
- AMBER (Wang et al., 2023) is tailored to assess both generative and discriminative tasks, including existence, attribute, and relation hallucination. We follow the $AMBER$ score in the original paper.
- MMHal-Bench (Sun et al., 2023) is composed of high-quality image-question pairs to measure the hallucination, and the generated responses are automatically rated by GPT-4.
- HallBench (Guan et al., 2024) is meticulously handcrafted by experienced human experts, and evaluated by a text-only GPT4-assisted evaluation framework.
- MMIU (Meng et al., 2024) is designed to assess abilities across diverse multi-image tasks, encompassing 7 types of multi-image relationships, and 11K meticulously curated multiple-choice questions.

Table 10: Two-stage training details. VLMs utilize various strategies to project vision tokens, such as MLP-based projectors.

| | | Stage One Pre-Training | Stage Two Instruction Tuning |
|---|---|---|---|
| Modules | Vision Encoder | Frozen | Frozen |
| | Projector | Trainable | Trainable |
| | Large Language Model | Frozen | Trainable |
| | Transformer Block (Ours) | Trainable | Trainable |
| Settings | Batch Size | 256 | 256 |
| | Learning Rate | - | 1e-4 |
| | MM Learning Rate | 5e-4 | 1e-5 |
| | Warmup Ratio | 0.05 | 0.02 |
| | Optimizer | AdamW (Loshchilov & Hutter, 2019) | AdamW (Loshchilov & Hutter, 2019) |
| | Epoch | 1 | 2 |

- MuirBench (Wang et al., 2024a) consists of 12 diverse multi-image tasks, utilizing a pairwise construction approach. Each standard instance is paired with a minimally semantically distinct unanswerable variant to ensure reliable assessment.

- MV-Bench (Li et al., 2024) covers 20 challenging video tasks intractable via single frames. Specifically, it transforms diverse static tasks into dynamic ones enables video tasks requiring a broad spectrum of temporal skills.

- Video-MME (Liu et al., 2024d) is the first full-spectrum benchmark in video analysis, distinguished by diverse video coverage, comprehensive temporal scope, multi-modal integration, and expert manual annotations, ensuring precise, reliable model assessment.

**Evaluations.** For the eight selected benchmarks, we largely follow their original evaluation metrics. Besides, to assess the level of hallucination snowballing in multi-agent contexts, we propose a hallucination snowballing score (*HS*), which quantifies both the severity and propagation of hallucinations. The score could be formulated as follows:

$$HS = \frac{1}{N} \sum_{i=1}^{N} \frac{1}{1 + \exp(\frac{D}{2} - d_i)} h_i, \tag{7}$$

where $d$ is the hallucination propagation distance, $D$ and $N$ are the total distance and the number of agents in the multi-agent system. $h$ represents the severity of hallucination, a centesimal-point scale score produced by a judge model. Here, we employ GPT-5-20250807 as the judge with the prompt as shown in the end of this paper. The average score on benchmarks measures the hallucination snowballing; the deeper and broader the snowballing of the hallucination, the higher the score.

To investigate the sensitivity of hallucination severity assessment, based on external judges, we conduct comparative experiments employing Gemini 2.5 Pro (Comanici et al., 2025) as the judge based on POPE benchmark. We observe that over 94% of data discrepancies fall below 10%, demonstrating the robustness of judge strategy. In practical applications, it suffices to ensure all comparisons are conducted under identical judge settings to guarantee the fairness of evaluations.

**Implementations.** Experiments are conducted on four or eight NVIDIA H20 96G GPUs. The salience of unimodal morphology $\omega$ is 0.3, the temperature scaling $\tau$ is 0.8, the reallocation coefficient $\alpha_1, \alpha_2$ in the middle and deep layers are set to 0.1 and 0.3, and the temperature of generation is set to 1 for MAS. For other configurations of baselines, we refer to the original paper.

**Training Pipelines.** Our proposed method can be integrated with other base VLMs in MAS to alleviate the multi-agent hallucination snowballing, requiring only one additional module for visual relay selection. Following the typical training paradigm, we employ a two-stage training process including a pre-training stage and instruction tuning stage, as reported in Table 10.

### D.2 ADDITIONAL RESULTS

**Additional Analyses of Visual Relay Tokens.** To further validate the effectiveness of our selected visual relay tokens in the visual flow, we conduct additional analyses with transformed visual features and different combinations of subsets of vision tokens. Specifically, we employ an average

pooling, a two-layer MLP, and a lightweight transformer (Mehta et al., 2021) for visual token compression, respectively. We also adopt object-level visual features (Neo et al., 2025), wherein we utilize an external segmentation model (i.e., SAM2) and incorporate the vision tokens of the predicted mask to relay visual information. Furthermore, we compare the results of different combinations of vision token subsets, as defined in Table 1. These additional tokens are randomly selected from the subsets, with the number of additional selections maintained below that of unimodal tokens.

As presented in Table 11, uniformly compressed visual features fail to relay information among agents and even exacerbate visual hallucinations due to vision transformation loss, particularly in complex visual scenarios (*e.g.*, in MMHal-Bench and HallBench benchmarks). Object-level visual features also yield suboptimal performance, owing to obstacles in external information selection and the introduction of additional latency. Additionally, unimodal vision tokens, adopted for visual information relay, do not gain significant benefits from the integration of other vision token subsets, while incurring extra time and computational overheads.

**Correction Capability on Adversarial Visual Inputs.** We assess the correction ability of our ViF in adversarial and noisy scenarios, including injecting edited images and mismatched images. The former randomly masks the area in the image, and the latter directly inputs mismatched and wrong image, both of them serve as strong adversarial scenarios. We stochastically inject the adversarial image in the 2 to 4 agent turns, and assess the performance in the following 5, 10, 15, and 20 agent turn, respectively.

As depicted in Figure 10, our ViF exhibits superior correction capability over the baseline when processing noisy and adversarial visual inputs, achieving by dynamically revising visual cognition across agent turns rather than adhering rigidly to prior outputs. As mentioned earlier, base VLMs in the multi-agent context tend to over-rely on prior texts to relay erroneous visual information. Conversely, our proposed visual flow mitigates the propagation and snowballing of hallucinations, thereby enabling enhanced correction and anti-adversarial capacities.

**Combination with Other Hallucination Mitigation Strategies.** To further enhance the compatibility and applicability of our method, we evaluate the performance of combinations with existing hallucination mitigation strategies, namely MemVR (Zou et al., 2025), VISTA (Li et al., 2025c), FarSight (Tang et al., 2025b), DeCo (Wang et al., 2025), and TAME (Tang et al., 2025a). As compared in Figure 11, we observe that most strategies achieve further improvements when combined with our ViF in multi-agent environments.

**Hyper-Parameter Analyses.** There are three key hyper-parameters in our proposed method, *i.e.*, the salience of unimodal morphology $\omega$ when selecting visual relay tokens with unimodal distribution, the temperature scaling $\tau$ in Equation 2, and the reallocation coefficient $\alpha_1$ and $\alpha_2$ in Equation 3 of the middle and deep layers. As listed in Table 12, the lower the $\omega$, the more proportions of visual tokens would be included; however, excessive visual relay tokens will not bring extra performance improvement but computation costs. When $\omega$ is set to 0.3, the model obtains the best results with relatively less token overhead. Besides, as shown in Table 13, and Table 14, when $\tau$, $\alpha_1$, $\alpha_2$ are set to 0.8, 0.1, 0.3, our model exhibits the greatest potential.

**Efficiency Analyses.** As reported in Table 8, our proposed ViF incurs only marginal additional computational overhead in respect of average latency and average number of operations. Additionally, as listed in Table 15, the computational overhead of our method remains relatively constant and exhibits no substantial increase with higher resolutions.

**Case Study.** As demonstrated in Figure 12, we visualize the generation procedure of the MAS equipped with our proposed ViF on four selected samples from two benchmarks. We observe that our method effectively mitigates the snowballing of multi-agent visual hallucinations, thereby enhancing overall performance. As shown in Example (b), although the agent outputs incorrect answers regarding object detection in the first turn, subsequent turns still accurately identify the perceptual target through visual flow information. Furthermore, as illustrated in Examples (c) and (d), misunderstandings of visual information in images lead to erroneous semantic outputs in early agent turns; however, such errors are not propagated throughout the multi-agent procedure via the visual flow, thus suppressing the snowballing of visual hallucinations.

Table 11: Results of different transformed visual features or various combinations of subsets of vision tokens on LLaVA-NeXT-7B and circular structure, as the visual flow to relay visual information. Due to the need for external model to obtain object-level visual features, we cannot calculate the end-to-end latency of this method. "Perf." indicates the quantitative performance on the benchmark.

| Visual Flow | CHAIR Perf.↓ | Latency↓ | POPE Perf.↑ | Latency↓ | AMBER Perf.↑ | Latency↓ | MMHal-Bench Perf.↑ | Latency↓ | HallBench Perf.↑ | Latency↓ |
|---|---|---|---|---|---|---|---|---|---|---|
| Baseline | 43.0 | **3.16** | 91.0 | **2.46** | 89.4 | **2.79** | 47.9 | **3.48** | 53.1 | **3.91** |
| Compressed Visual Features (Pooling) | 43.9 | 3.33 | 89.6 | 2.61 | 89.2 | 2.95 | 42.5 | 3.64 | 46.7 | 4.05 |
| Compressed Visual Features (MLP) | 42.7 | 3.40 | 91.6 | 2.73 | 90.4 | 3.02 | 44.9 | 3.77 | 51.8 | 4.18 |
| Compressed Visual Features (Transformer) | 42.2 | 3.61 | 91.9 | 2.96 | 91.1 | 3.27 | 48.5 | 3.99 | 53.3 | 4.40 |
| Object-level Visual Features | 42.5 | - | 92.3 | - | 91.8 | - | 47.2 | - | 53.4 | - |
| (e) Unimodal + (a) Rise | **41.0** | 3.55 | 92.7 | 2.87 | 92.1 | 3.20 | 50.6 | 3.87 | 54.8 | 4.33 |
| (e) Unimodal + (b) Down | 42.0 | 3.56 | 92.3 | 2.89 | 91.3 | 3.22 | 49.7 | 3.88 | 53.7 | 4.34 |
| (e) Unimodal + (a) Rise + (b) Down | 41.3 | 3.63 | 93.1 | 2.94 | **92.9** | 3.28 | 50.3 | 3.94 | 54.1 | 4.40 |
| **(e) Unimodal (Ours)** | 41.2 | 3.47 | **93.3** | 2.79 | 92.7 | 3.10 | **51.1** | 3.83 | **55.7** | 4.23 |

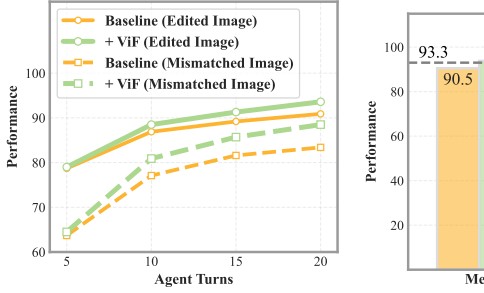
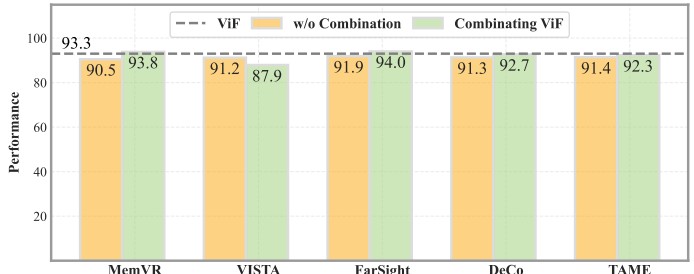

Figure 10: Analyses of correction ability on LLaVA-Next-7B and circular structure when feeding adversarial visual inputs, evaluated by POPE benchmark.

Figure 11: Comparisons of the combination of our proposed ViF and other hallucination mitigation strategies on LLaVA-Next-7B and circular structure, evaluated by POPE benchmark. Specifically, we incorporate five training-free methods, which are seamlessly integrated with our approach.

Table 12: Influence of the salience of unimodal morphology $\omega$ on LLaVA-NeXT-7B and circular MAS structure.

| $\omega$ | Ratio % | CHAIR↓ | POPE↑ | AMBER↑ | MMHal-Bench↑ | Hall Bench↑ |
|---|---|---|---|---|---|---|
| 0.1 | 17.6 | 44.5 | 90.2 | 88.6 | 45.9 | 52.7 |
| 0.2 | 6.7 | 42.8 | 91.5 | 89.2 | 49.1 | 52.8 |
| 0.3 | 2.3 | **41.2** | **93.3** | **92.7** | **51.1** | **55.7** |
| 0.4 | 1.3 | 41.7 | 92.5 | 92.0 | 49.8 | 54.9 |
| 0.5 | 0.2 | 42.9 | 91.1 | 89.6 | 47.9 | 53.0 |

Table 13: Influence of the temperature scaling $\tau$ on LLaVA-NeXT-7B and circular MAS structure.

| $\tau$ | CHAIR↓ | POPE↑ | AMBER↑ | MMHal-Bench↑ | Hall Bench↑ |
|---|---|---|---|---|---|
| 0.6 | 44.1 | 91.4 | 90.8 | 48.2 | 52.1 |
| 0.7 | 43.0 | 92.4 | 91.8 | 50.3 | 54.2 |
| 0.8 | 41.2 | **93.3** | **92.7** | **51.1** | **55.7** |
| 0.9 | **41.0** | 93.1 | 92.2 | 48.9 | 53.8 |
| 1.0 | 41.7 | 92.5 | 91.6 | 47.7 | 52.8 |

Table 14: Influence of the reallocation coefficient $\alpha_1$ and $\alpha_2$ on LLaVA-NeXT-7B and circular MAS structure.

| $\alpha_1$ | $\alpha_2$ | CHAIR↓ | POPE↑ | AMBER↑ | MMHal-Bench↑ | Hall Bench↑ |
|---|---|---|---|---|---|---|
| 0.0 | 0.2 | 43.2 | 89.6 | 89.1 | 47.2 | 47.6 |
| 0.0 | 0.3 | 42.8 | 90.0 | 89.3 | 47.4 | 48.2 |
| 0.0 | 0.4 | 43.0 | 89.8 | 89.0 | 47.1 | 47.8 |
| 0.1 | 0.2 | 41.3 | 93.1 | 92.6 | 50.7 | 55.3 |
| 0.1 | 0.3 | **41.2** | **93.3** | **92.7** | **51.1** | **55.7** |
| 0.1 | 0.4 | 41.8 | 92.9 | 92.4 | 50.5 | 54.9 |
| 0.2 | 0.2 | 42.9 | 90.6 | 89.2 | 47.3 | 48.1 |
| 0.2 | 0.3 | 43.1 | 89.8 | 88.5 | 46.9 | 48.0 |
| 0.2 | 0.4 | 42.9 | 89.6 | 88.9 | 46.5 | 47.7 |

Table 15: Efficiency comparison between our ViF and the base models on LLaVA-NeXT-7B and the circular MAS architecture, which are evaluated with different image resolutions. We employ the original-resolution samples from the CHAIR benchmark and conduct bilinear interpolation to downsample them to 1/2, 1/4, and 1/8 of the initial resolution.

| Base Agent | 1/8 Resolution | | 1/4 Resolution | | 1/2 Resolution | | Original Resolution | |
|---|---|---|---|---|---|---|---|---|
| | *Latency↓* | *FLOPs↓* | *Latency↓* | *FLOPs↓* | *Latency↓* | *FLOPs↓* | *Latency↓* | *FLOPs↓* |
| Baseline | 1.90 | 103.5 | 2.27 | 120.8 | 2.69 | 133.6 | 3.16 | 157.3 |
| **+Ours** | 2.17 | 112.2 | 2.57 | 130.0 | 3.01 | 143.8 | 3.47 | 168.5 |

**Example (a)**

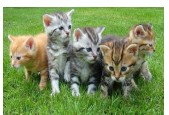

Are all of the cats the same color?

**Agent Turn 1:** There are many cats and they are in different colors.
...
**Agent Turn 2:** One is orange, two are grey, and two are mixed ones, so they are not the same colors.
...
**Agent Turn 3:** It is confirmed that their coats are diverse, so the answer is no.

**Example (b)**

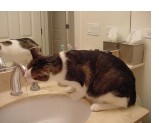

Is there a toothbrush in the image?

**Agent Turn 1:** There is a cat on the sink, faucets, a tissue holder, and a mirror. But there is no toothbrush. So the answer is no.
...
**Agent Turn 2:** There are some small pink and green brush-shaped objects, typically found in toothbrush cups beside sinks. So the answer is yes.
...
**Agent Turn 3:** Those upright, rod-like objects matched the shape and position of toothbrushes perfectly. So the answer is yes.

**Example (c)**

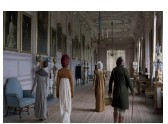

Do the persons live in this house? Answer it and give the rationale.

**Agent Turn 1:** These people were dressed with great refinement, and the man on the right holds a sword in his hand, possibly a knight of noble birth. The decor of the room was exceptionally lavish, perfectly matching their attire. So they are likely live in the house.
...
**Agent Turn 2:** The corridor was arranged with great luxury and solemnity. And these people walked straight ahead, with general clothing, as if they were on a tour or being led, rather than moving about in a familiar house. So the people probably don't live here.
...
**Agent Turn 3:** Based on the way these people are dressed and also their behavior and demeanor, they are more like tourists. So they don't live here.

**Example (d)**

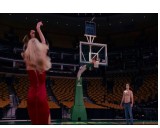

Why does the woman have both of her hands in the air? Answer it and give the rationale.

**Agent Turn 1:** The woman stands on a basketball court with a hoop in front of her, but there is no basketball. She is dressed gorgeously and swinging her arms, she is likely dancing in the court.
...
**Agent Turn 2:** She shot a basketball because raising both hands is crucial for generating power and accuracy in a jump shot or set shot.
...
**Agent Turn 3:** She just shot the basketball because she is in the basketball context and the movements of her arms indicate a shoot.

Figure 12: Case study of the results of ViF based on LLaVA-NeXT-7B across MM-Vet and POPE benchmarks. For clarity, we choose the linear structure and the agent turn is set to 3.

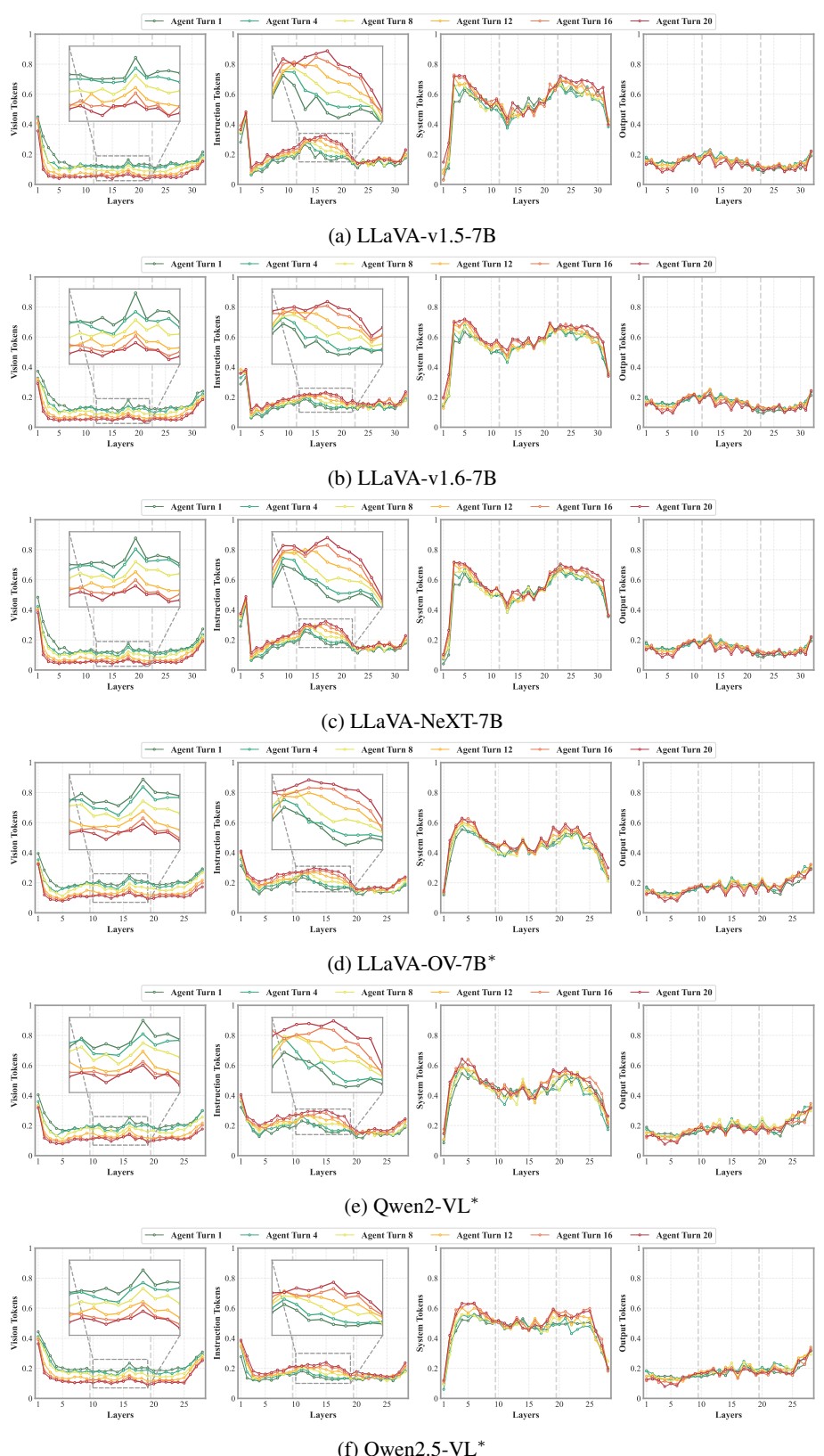

Figure 13: Layer-wise attention allocation of six models in different agent turns, and ∗ denotes that using Key-Norm to replace the attention score.

Table 16: Results of six VLMs when dropping different selected subsets of vision token in the shallow, middle, and deep layers.

| | | Shallow Layers | | | | Middle Layers | | | | Deep Layers | | | |
|---|---|---|---|---|---|---|---|---|---|---|---|---|---|
| | | 25% | 50% | 75% | 100% | 25% | 50% | 75% | 100% | 25% | 50% | 75% | 100% |
| LLaVA-v1.5-7B | *w/o* Dropping | | | | | | 83.0 | | | | | | |
| | (a) Random | 49.1 | 41.3 | 36.1 | 28.7 | 77.3 | 64.1 | 60.2 | 56.2 | 81.6 | 81.1 | 80.6 | 80.2 |
| | (b) Inactive | 53.3 | 44.6 | 38.9 | 29.7 | 82.7 | 78.9 | 80.1 | 76.7 | 82.8 | 81.9 | 82.1 | 81.7 |
| | (c) Rise | 40.1 | **33.1** | **27.5** | **18.4** | 76.4 | 62.0 | 54.5 | 50.8 | **81.0** | **79.3** | **79.8** | **80.0** |
| | (d) Fall | 40.0 | 36.4 | 28.2 | 19.9 | 75.2 | 62.5 | 55.6 | 50.0 | 82.4 | 81.6 | 81.4 | 80.6 |
| | **(e) Unimodal** | **39.6** | 35.9 | 27.9 | 22.4 | **51.3** | **43.1** | **35.0** | **22.6** | 81.6 | 81.3 | 80.5 | 80.9 |
| LLaVA-v1.6-7B | *w/o* Dropping | | | | | | 83.3 | | | | | | |
| | (a) Random | 49.9 | 42.2 | 37.2 | 28.0 | 77.0 | 63.6 | 60.1 | 58.1 | 81.8 | 81.1 | 80.7 | 80.9 |
| | (b) Inactive | 52.1 | 43.8 | 39.6 | 30.5 | 83.6 | 82.4 | 78.6 | 76.1 | 82.9 | 83.1 | 82.7 | 82.6 |
| | (c) Rise | **39.2** | **33.8** | **28.2** | **18.5** | 78.9 | 61.6 | 54.7 | 51.2 | **81.4** | **80.8** | **80.4** | **80.1** |
| | (d) Fall | 39.6 | 36.6 | 28.7 | 20.2 | 75.8 | 62.9 | 57.5 | 50.3 | 82.5 | 82.4 | 81.0 | 81.2 |
| | **(e) Unimodal** | 40.9 | 36.4 | 28.0 | 20.9 | **46.0** | **42.9** | **33.9** | **23.4** | 82.4 | 81.6 | 80.6 | 80.1 |
| LLaVA-NeXT-7B | *w/o* Dropping | | | | | | 85.2 | | | | | | |
| | (a) Random | 51.8 | 44.5 | 38.4 | 30.5 | 78.9 | 66.1 | 62.7 | 59.0 | 84.4 | 83.2 | 82.9 | 82.6 |
| | (b) Inactive | 55.1 | 46.2 | 41.8 | 32.5 | 84.3 | 82.9 | 81.5 | 78.3 | 85.0 | 84.6 | 84.3 | 84.6 |
| | (c) Rise | 41.9 | **35.6** | **29.6** | **20.8** | 79.4 | 64.2 | 56.4 | 52.3 | **83.6** | **82.7** | **81.8** | **81.6** |
| | (d) Fall | **41.6** | 38.8 | 30.7 | 22.5 | 78.3 | 64.8 | 58.5 | 52.9 | 84.1 | 82.8 | 82.0 | 82.4 |
| | **(e) Unimodal** | 42.1 | 37.6 | 30.0 | 22.8 | **52.9** | **44.5** | **36.6** | **25.3** | 84.4 | 83.0 | 82.3 | 81.8 |
| LLaVA-OneVision-7B | *w/o* Dropping | | | | | | 87.6 | | | | | | |
| | (a) Random | 52.7 | 45.1 | 41.4 | 33.2 | 82.4 | 72.5 | 66.8 | 63.7 | 87.5 | 86.6 | **86.2** | 86.5 |
| | (b) Inactive | 58.8 | 46.5 | 44.9 | 36.4 | 83.6 | 80.2 | 74.4 | 67.8 | 87.7 | 87.3 | 87.2 | 87.0 |
| | (c) Rise | 44.7 | **39.5** | **31.1** | **22.4** | 81.2 | 65.4 | 62.4 | 55.5 | 87.1 | 86.8 | 86.7 | 86.5 |
| | (d) Fall | 45.2 | 42.1 | 33.9 | 24.8 | 82.6 | 66.1 | 59.6 | 53.6 | 87.2 | 86.7 | 86.6 | **86.4** |
| | **(e) Unimodal** | **44.6** | 41.7 | 33.8 | 26.1 | **53.0** | **46.2** | **39.7** | **27.5** | **86.8** | **86.3** | 86.5 | 85.7 |
| Qwen2-VL-7B | *w/o* Dropping | | | | | | 85.9 | | | | | | |
| | (a) Random | 53.2 | 44.7 | 39.8 | 31.2 | 79.3 | 67.1 | 62.6 | 59.0 | 84.8 | **84.0** | **83.5** | **83.0** |
| | (b) Inactive | 55.5 | 46.5 | 42.0 | 33.5 | 86.4 | 83.1 | 83.2 | 78.4 | 85.6 | 85.4 | 85.2 | 85.7 |
| | (c) Rise | 43.2 | **36.7** | **29.8** | **21.2** | 80.0 | 64.2 | 56.5 | 53.9 | **85.2** | 84.8 | 84.1 | 83.7 |
| | (d) Fall | **43.0** | 40.1 | 31.6 | 23.4 | 78.9 | 65.5 | 60.0 | 52.8 | 84.4 | 84.2 | 84.6 | 83.9 |
| | **(e) Unimodal** | 44.2 | 39.0 | 31.0 | 23.5 | **53.8** | **46.0** | **37.1** | **25.7** | 84.5 | 84.0 | 83.9 | 83.4 |
| Qwen2.5-VL-7B | *w/o* Dropping | | | | | | 85.6 | | | | | | |
| | (a) Random | 51.5 | 44.8 | 39.2 | 31.2 | 80.0 | 66.3 | 63.9 | 59.5 | **84.4** | 83.9 | 83.6 | 83.4 |
| | (b) Inactive | 56.5 | 46.3 | 41.9 | 33.2 | 83.7 | 82.2 | 80.9 | 78.2 | 84.8 | 84.6 | 84.7 | 84.5 |
| | (c) Rise | 42.4 | **36.6** | **30.4** | **20.9** | 80.3 | 63.6 | 55.7 | 52.5 | 84.9 | **83.3** | **83.2** | 83.1 |
| | (d) Fall | 42.7 | 38.8 | 31.0 | 23.0 | 80.0 | 64.2 | 59.0 | 54.3 | 84.8 | 83.9 | 83.3 | **82.8** |
| | **(e) Unimodal** | **42.1** | 38.1 | 30.4 | 23.1 | **54.0** | **44.7** | **37.0** | **26.0** | 85.0 | 84.8 | 83.6 | 83.4 |

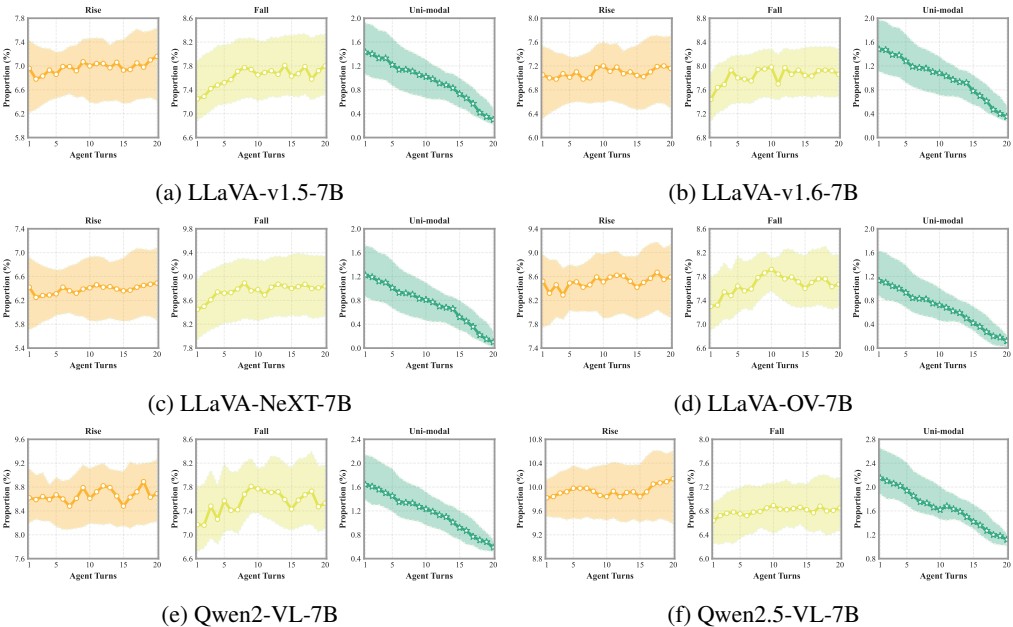

(a) LLaVA-v1.5-7B  (b) LLaVA-v1.6-7B

(c) LLaVA-NeXT-7B  (d) LLaVA-OV-7B

(e) Qwen2-VL-7B  (f) Qwen2.5-VL-7B

Figure 14: Proportion of vision tokens subsets of six models in different agent turns.

Prompt for Evaluating the Severity of Hallucination $h$

You are a strict visual hallucination judge for vision-language models.

Your job is to evaluate the level of visual hallucination, given:
    - 1. The user instruction and corresponding visual inputs.
    - 2. The full generated output from the model.
    - 3. The ground-truth of the instruction.
You must detect hallucinations and rate their severity on a 0 to 100 scale.

Definitions:
    - A "visual hallucination" is: a class of hallucination phenomena in which a vision
        -language model, conditioned on real visual inputs, generates visual details
        in tasks that are inconsistent with, incorrect about, or entirely fabricated
        beyond the given visual content, thereby making its outputs unfaithful to the
        visual evidence. We care about hallucinations in both facts and reasoning.

Your tasks:
    - 1. Decide whether the generated output contains visual hallucinations. (combining
         visual inputs and ground-truth for verification)
    - 2. If yes, briefly explain why they are hallucinations.
    - 3. Output a SEVERITY score from 0-100 (integer).

Severity guidelines (0-100):
    - 0: No hallucination. Fully consistent with references and context.
    - 1-20: Very minor issues, local details, or small inaccuracies that do not change
        the main conclusion.
    - 21-40: Clear but localized hallucinations. The main conclusion is still mostly
        correct.
    - 41-60: Important hallucinations that significantly affect part of the answer or
        core reasoning.
    - 61-80: Severe hallucinations. The answer is largely incorrect or misleading.
    - 81-100: Extreme hallucinations. The answer is almost entirely fabricated or
        contradicts the references.

Output format:
- You MUST output a single valid JSON object with these fields:
  - "explanation": short natural language explanation (1-3 sentences, can be empty if
      there is no hallucination)
  - "severity": integer in [0, 100]

Be concise but precise in your explanation. Do NOT include any text outside the JSON.

