# OpenReview forum: "Visual Multi-Agent System: Mitigating Hallucination Snowballing via Visual Flow"
_ICLR.cc/2026/Conference — ICLR 2026 Poster_

### Official Review · Reviewer_7MHV · 2025-10-16

**Soundness:** 3
**Presentation:** 4
**Contribution:** 3
**Rating:** 6
**Confidence:** 4

**Summary:**

The authors propose ViF, a lightweight plug-and-play method to reduce visual hallucination snowballing in Visual Multi-Agent Systems built on Vision-Language Models. They observe that as multi-agent interactions progress, agents increasingly rely on textual context while losing attention to visual evidence, causing errors to compound. To address this, ViF introduces two mechanisms: Visual Information Relay, which passes key visual tokens between agents to preserve visual grounding, and Attention Reallocation, which restores attention to visual features in middle and deep layers. Since modern VLMs use Flash-Attention, which hides attention maps, ViF uses the key norm to identify important vision tokens efficiently. Experiments on eight benchmarks and ten base models show consistent gains of about 2–3.5% in both general and hallucination-focused evaluations. Overall, ViF is an effective and easily integrable framework that improves visual consistency and reduces hallucination propagation without retraining underlying models.

**Strengths:**

1. The proposed ViF method is lightweight and plug-and-play, requiring no model retraining or architecture modification, making it highly practical for integration into existing vision-language frameworks.

2. The paper provides clear empirical evidence through attention analysis that supports the identified problem and motivates the design of ViF, showing strong interpretability and reasoning behind the approach.

3. Extensive experiments across eight benchmarks and ten base models demonstrate the method’s robustness and generality, with consistent performance gains across both general and hallucination-specific evaluations.

4. The work contributes to the reliability and stability of multi-agent VLM systems, an emerging but underexplored research direction, making it a meaningful step toward safer and more grounded visual reasoning in collaborative AI agents.

**Weaknesses:**

1. Although the paper claims ViF reduces hallucination propagation, it does not include qualitative or human evaluations to confirm whether the mitigated errors are genuinely perceptual or semantic, limiting the interpretability of improvements.

2. The key-norm heuristic used as a substitute for attention may not perfectly capture true token importance, and the paper does not analyze failure cases or potential mismatches between key-norm values and actual visual relevance.

3. The attention reallocation strategy is presented as effective, but the specific reallocation rule (e.g., scaling factors or redistribution formula) is not fully described, reducing reproducibility.

4. The framework assumes that hallucination arises primarily from attention degradation, but other causes of hallucination are not addressed, which may limit the generality of the approach.

5. The computational overhead and latency introduced by token relaying between agents are not quantified, which could be important for real-time or large-scale multi-agent applications.

**Questions:**

1. How were the attention reallocation parameters determined? Are they fixed hyperparameters or adaptively tuned per model and layer?

2. When using key-norm as a proxy for attention importance, did you compare it empirically with models where attention maps are accessible to verify its reliability?

3. Could you provide qualitative examples showing how ViF changes an agent’s reasoning trajectory or visual focus compared to the baseline?

4. Since ViF passes selected vision tokens between agents, how do you prevent error propagation if the selected tokens themselves are noisy or misleading?

5. Have you tested whether combining ViF with other hallucination mitigation techniques (e.g., grounding via retrieval or contrastive filtering) yields additive improvements?

6. What is the computational cost (both in FLOPs and latency) introduced by ViF during inference, and is it negligible compared to the base MAS pipeline?

---

> ### Author Response · Authors · 2025-11-21
> **Response to Reviewer 7MHV [1/2]**
>
> We greatly appreciate your insightful and detailed feedback. Your comments have been carefully discussed and fully taken into account, and we improve our work accordingly.
>
> ### **Addressing Weaknesses**
> >**1. Interpretability**:
>
> To improve interpretability, we provide four examples as case study in **Figure 12**. More precisely, as shown in **Example (b)**, although the agent outputs incorrect answers regarding object detection in the first turn, subsequent turns still accurately identify the perceptual target through visual flow information. Furthermore, as illustrated in **Examples (c) and (d)**, misunderstandings of visual information in images lead to erroneous semantic outputs in early agent turns. However, such errors are not propagated throughout the multi-agent procedure via the visual flow, thus suppressing the snowballing of visual hallucinations.
>
>
> >**2. Supplement to Key-Norm Based Strategy**:
>
> Thank you for pointing out potential ambiguity. Due to the length constraints of the main text, we only provide a brief introduction to this strategy, but we elaborate on it comprehensively in **Appendix C.2** in our last manuscript. Specifically, we conduct systematic comparisons between the key-norm based strategy and the original attention based strategy: as detailed in **Table 9**, quantitative evaluations across eight benchmarks show that when using 3 buffer tokens, the key-norm based strategy almost reaches the full performance scores of the attention based method and even outperforms it on three benchmarks. In addition, as visualized in **Figure 9**, our qualitative assessments of the selected tokens from the two strategies indicate that the token selection of the key-norm based method (with 3 buffer tokens) is closely similar to that of the attention score based strategy. Although there are inevitable minor differences between the two strategies, based on the comprehensive support from qualitative and quantitative comparisons, we confirm that the key-norm substitute is both effective and robust.
>
>
> >**3. Supplement to Attention Reallocation**:
>
> Thank you for pointing out potential ambiguity. The details of attention reallocation are specified in **Section 3.2**, where we  elaborate on the entire reallocation process through **Equations 2, 3, and 4** and add supplementary explanations. We provide a more detailed explanation of the notations and meanings of each component in the formula to improve clarity. And for all hyper-parameters used in our work, we not only provide its specific values in **Appendix D.1** but also conduct a sensitivity analysis to verify its impact, as summarized in **Table 12,13,14**.
>
>
> >**4. Other Causes of Hallucination**:
>
> We sincerely understand your concern that hallucinations can be attributed to other causes. However, we would like to clarify that the main focus of our work is specifically on visual hallucination snowballing in multi-agent environments. The analyses in **Section 2** and the supporting results in **Table 6** strongly validate this research focus, indicating that existing mitigation methods tailored for sing-agent fail to address the hallucination snowballing in MAS. Therefore, our work primarily targets the degradation of visual attention as the core issue underlying the hallucination snowballing phenomenon.
>
>
>
> >**5. Latency and Computational Overload**:
>
> We report the latency and computational overhead of our method. As presented **below** (complete in **Table 8**), our ViF demonstrates high efficiency with moderate overheads, incurring an additional 8.1–13.4\% inference latency and 4.8–11.9\% computational overhead (measured by FLOPs) compared to the base model across five selected benchmarks. Additionally, our analyses of latency and computational overhead with respect to base model scale and visual input resolution further believe the efficiency and practicality of ViF in real-time or large-scale multi-agent applications.
>
>
> | | CHAIR | | POPE | | AMBER | | MMHal-Bench | | HallBench | |
> |--|--|--|--|--|--|--|--|--|--|--|
> | | Latency↓ | FLOPs↓ | Latency↓ | FLOPs↓ | Latency↓ | FLOPs↓ | Latency↓ | FLOPs↓ | Latency↓ | FLOPs↓ |
> | LLaVA-NeXT-7B | 3.16 | 157.3 | 2.46 | 103.4 | 2.79 | 127.2 | 3.48 | 184.0 | 3.91 | 248.3 |
> | +Ours | 3.47 | 168.5 | 2.79 | 115.7 | 3.10 | 138.9 | 3.83 | 197.6 | 4.23 | 260.3 |

---

> ### Author Response · Authors · 2025-11-21
> **Response to Reviewer 7MHV [2/2]**
>
> ### **Response to Questions**
> >**1. Parameters of Attention Reallocation**:
>
> To clarify, in the attention reallocation procedure, both $\tau$ that amplifies dynamic trends and reallocation coefficient $\alpha$ that controls the degree of attention reallocation are determined through experimental analyses. Specifically, $\tau=0.8,\alpha_1=0.1,\alpha_2=0.3$, where $\alpha_1$ and $\alpha_2$ are the reallocation coefficients for the reallocation in the middle and deep layers, respectively. As detailed in **Table 12,13,14**, we conduct comparisons across different parameter settings and ultimately select the configuration that achieves the best comprehensive performance. Notably, these parameters are universally applicable to all models. Meanwhile, specialized parameters tailored to each individual model are also practicable if required.
>
>
> >**2. Supplement to Key-Norm Strategy**:
>
> As illustrated in **Figure 9**, we compare the selected subset of vision tokens between the original attention score based method and also the key-norm based method, which illustrates that the two selections are largely similar. Besides, the quantitative comparison in **Table 9** also verify its reliability.
>
>
>
> >**3. Qualitative Examples**:
>
> We provide four qualitative examples in **Figure 12**, demonstrating the influence of ViF to mitigate hallucination snowballing. As described in the response **Weakness 2**, it is observed that ViF prevent initial
> hallucinations to be propagated in the MAS, which will propagate and snowball in the baselines.
>
>
> >**4. Correction in Adversarial Scenarios**:
>
> To evaluate performance under noisy or misleading visual inputs, we compare our ViF with the baseline under the condition of inputting edited (randomly masked) or mismatched (from other questions) images across multi-agent turns. Specifically, we randomly inject the adversarial images across the first five turns, and evaluate their performance in the following turns (Agent Turn 5, 10, 15, 20). As presented **below** (complete in **Figure 10**), despite the input of noisy or misleading visual information, ViF exhibits excellent correction capacity and robustness, surpassing the baseline by 2.7–5.1\%. This indicates that the baselines tend to over-rely on prior texts to transmit erroneous visual information; in contrast, our visual flow alleviates hallucination propagation and snowballing, achieving superior corrective and anti-adversarial performance.
>
> | Inputs | Method | Agent Turn 5 | Agent Turn 10 | Agent Turn 15 | Agent Turn 20 |
> |--|--|--|--|--|--|
> | Edit  Image   | Baseline | 78.7 | 86.9 | 89.2 | 90.9 |
> | Edit  Image   | +Ours   | **79.0** | **88.5** | **91.3** | **93.6** |
> | Mismatched Image   | Baseline | 63.7 | 77.1 | 81.6 | 83.4 |
> | Mismatched Image   | +Ours | **64.5** | **80.9** | **85.7** | **88.5** |
>
>
> >**5. Combination with Other Hallucination Mitigation Strategies**:
>
> We evaluate the performance of our ViF combined with five existing hallucination mitigation strategies. As presented in **Figure 11**, most strategies achieve further gains when integrated with ViF in multi-agent settings. We intend to perform further experiments and technical optimizations to improve its compatibility with other strategies.
>
>
> |       | MemVR    | VISTA   | FarSight    | DeCo    | TAME    |
> |-------|------|------|------|------|------|
> | Baseline | 90.5 | **91.2** | 91.9 | 91.3 | 91.4 |
> | +Ours | **93.8** | 87.9 | **94.0** | **92.7** | **92.3** |
>
>
> >**6. Latency and Computational Overload**:
>
> As we mentioned in the response to **Weakness 5**, our ViF brings an additional 8.1–13.4\% inference latency and 4.8–11.9\% FLOPs across five selected benchmarks, which is relatively negligible compared to the base MAS pipeline.

---

> > ### Comment · Reviewer_7MHV · 2025-11-21
> >
> > Thank you for your comments. All of my concerns and questions have been fully addressed. I raised my score to 8.

---

### Official Review · Reviewer_HX6A · 2025-10-29

**Soundness:** 4
**Presentation:** 3
**Contribution:** 3
**Rating:** 6
**Confidence:** 5

**Summary:**

The critical failure of multi-agent systems lies in the compression and selective amplification of visual evidence by the "text stream", causing early hallucinations to spread along the information chain;ViF complements and stably transfers visual evidence through the "visual stream", and, in conjunction with attention redistribution, maintains visual dominance in the middle layer, significantly reducing the snowball effect in multi-round scenarios.

**Strengths:**

- **The problem definition is specific and MAS-oriented:** The paper identifies "visual hallucination snowballing" as a critical failure in multi-agent scenarios and makes a diagnosis from the triple perspectives of layer-round-token; this focus differs from the paradigm of traditional single-agent LVLM hallucination research.

- **The design concept of the visual stream is novel:** By combining "visual relay token + attention redistribution", a non-textual channel is introduced at the cross-agent communication level, aiming to stably retain and amplify visual evidence, and conceptually has a concise and scalable appeal.

- **Diagnosis and Ablation Adequacy:** Quantitative trends of inter-layer and round attention, hierarchical deletion of shallow/middle/deep layers, and the proportion of key subsets changing with rounds, with the overall analysis chain being complete; ablation covers the proportion of visual relay tokens and the contributions of each layer to attention redistribution, supporting the claim.

- **Compatibility Emphasis:** Positioned as "Plug and Play", it provides Key-Norm alternatives and buffer strategies on models with FlashAttention to lower the threshold for reproducibility.

**Weaknesses:**

1. **Computation and latency overhead not reported:** Transmitting visual relay tokens across agents increases sequence length and attention computation, and attention redistribution also introduces additional operators. It is recommended to report the end-to-end throughput/latency curves (varying with agent rounds, image resolution, model scale, and etc.), and compare them in parallel with the baseline of text-only streams.

2. **The coverage of the task domain is limited:** The main evaluation is on the understanding and question answering of static images. It is recommended to verify the generalization of ViF on agentic tasks with external tool usage, especially in multi-round planning and interaction scenarios, to examine whether visual relaying is more deeply coupled with the workflow.

3. **Reliability and External Validity of the HS Metric:** HS relies on the judge model to evaluate "hallucination severity" and "propagation depth". It is recommended to conduct  reviewer consistency and model sensitivity analysis , and perform external validity testing on the  manually annotated  snowball diagnostic set; at the same time, disclose the version and prompt of the judge model.

4. **Comparison with Structured Visual Features:** Besides "visual token direct transfer", schemes that transfering compressed/pooled spatial feature maps or object-level memory can be compared with ViF in terms of performance-overhead-robustness differences.

5. **Analysis of Strong Adversarial Scenarios:** When the visual judgment of the first agent is severely deviated or deliberately injected with errors, can ViF "correct rather than follow"? It is recommended to construct stronger adversarial distributions to evaluatethe cross-round correction rate of early errorsandthe anti-interference ability of the visual stream.

**Questions:**

1. **Cross-domain generalization:** Under video question answering, multi-image reasoning, and tool-enhanced workflows, is the benefit of ViF stable?

2. The proposed ViF approach seems to mainly focus on the impact of visual tokens that have undergone an **attention sink**. Previous studies have mainly focused on the level of single agent and advocated that alleviating the phenomenon of attention sink can improve the performance of the model.[1, 2] However, some researchers, like the author, have proposed using attention sink to enhance the performance of the model. Recent Qwen3-Next series model replaces standard attention with the combination of Gated DeltaNet and Gated Attention, which enables efficient context modeling for ultra-long context length while alleviates the attention sink. If the ViF method is used on VLMs developed based on such models in the future, can similar gains still be obtained?

ref:

[1] Huang Q, Dong X, Zhang P, et al. Opera: Alleviating hallucination in multi-modal large language models via over-trust penalty and retrospection-allocation[C]//Proceedings of the IEEE/CVF Conference on Computer Vision and Pattern Recognition. 2024: 13418-13427.

[2] Chen Z, Li M, Chen Z, et al. Advancing general multimodal capability of vision-language models with pyramid-descent visual position encoding[J]. arXiv preprint arXiv:2501.10967, 2025.

---

> ### Author Response · Authors · 2025-11-21
> **Response to Reviewer HX6A [1/2]**
>
> We greatly appreciate your insightful and detailed feedback. Your comments have been carefully discussed and fully taken into account, and we improve our work accordingly.
>
> ### **Addressing Weaknesses**
> >**1. Latency and Computational Overload**:
>
> We conduct experiments about the latency and computational overload (measured by FLOPs), comparing the performance on various models and settings. As reported **below** (complete in **Table 8**), the results confirm that our ViF maintains high efficiency with moderate overheads: across five selected benchmarks, it introduces an additional 8.1–13.4\% inference latency and 4.8–11.9\% FLOPs compared to the base model. Besides, results on 7B/13B/34B models indicate that the extra overhead remains stable across base models of larger scales, showing only a slight linear increase. Notably, for larger models (e.g., LLaVA-NeXT-34B), the additional latency and computation are even more negligible, falling below 4\% and 3\% respectively. Furthermore, as detailed in **Table 15**, the time and computational overhead of our ViF remain well-controlled and efficient when feeding visual images with different resolutions and across various agent turns.
>
> | | CHAIR | | POPE | | AMBER | | MMHal-Bench | | HallBench | |
> |--|--|--|--|--|--|--|--|--|--|--|
> | | Latency↓ | FLOPs↓ | Latency↓ | FLOPs↓ | Latency↓ | FLOPs↓ | Latency↓ | FLOPs↓ | Latency↓ | FLOPs↓ |
> | LLaVA-NeXT-7B | 3.16 | 157.3 | 2.46 | 103.4 | 2.79 | 127.2 | 3.48 | 184.0 | 3.91 | 248.3 |
> | +Ours | 3.47 | 168.5 | 2.79 | 115.7 | 3.10 | 138.9 | 3.83 | 197.6 | 4.23 | 260.3 |
>
>
> >**2. Performance on Augmented Visual Domains**:
>
> We fully agree that our ViF should be evaluated across more domains, including tool-enhanced capabilities. However, to the best of our knowledge, there is a lack of specialized tool-based benchmarks for evaluating VLMs, despite the existence of numerous benchmarks tailored for LLMs. As an alternative, we include augmented visual benchmarks, specifically for multi-image and video tasks. As presented **below** (complete in **Table 4**), our method achieves an average improvement of 2.0–4.9\% across the four base models on these benchmarks. If you are aware of any tool-based benchmarks for VLMs or wish to see results in other specific domains, please feel free to contact us at any point during the rebuttal period.
>
> | Base Agent       | MMIU↑   | MuirBench↑ | MVBench↑  | Video-MME↑ | Avg.↑  |
> |------------------|---------|------------|-----------|------------|--------|
> | LLaVA-NeXT-7B    | 31.6    | 42.6       | 49.2      | 60.4       |    |
> | +Ours            | 33.9 | 44.3 | 52.0 | 61.9 | 4.9   |
>
>
> >**3. Supplement to HS Metric**:
>
> Thank you for pointing out potential ambiguity. We now include necessary descriptions, the version and scoring prompt of the judge model utilized for hallucination score calculation. To evaluate the sensitivity of hallucination severity assessment, we perform comparative experiments with an alternative judge model. Notably, over 94\% of data discrepancies are below 10\%, verifying the robustness of our judge strategy. Although different judge models may still affect the assessment of the HS metric to a certain extent, consistent judge configurations across all comparisons guarantee evaluation fairness.

---

> ### Author Response · Authors · 2025-11-21
> **Response to Reviewer HX6A [2/2]**
>
> >**4. Comparison with Compressed/Object-level Visual Features**:
>
> We fully understand and appreciate your suggestion. Considering the potential perception loss and redundancy of compressed or object-level visual features, we believe that our selected unimodal vision tokens are more suitable to relay visual information accurately. To further prove our view, we supplement four categories of visual features for comparison: 1) Compressed Visual Features (Pooling), 2) Compressed Visual Features (MLP), 3) Compressed Visual Features (Transformer), and 4) Object-level Visual Features. For the first three types, we employ an average pooling, a two-layer MLP, and a lightweight transformer respectively for visual token compression; for the last one, objects are previously segmented by external model. As report **below** (complete in **Table 11**), the results show that these uniformly compressed visual features are unable to effectively relay information among agents and even exacerbate visual hallucinations, 1.8-5.5\% behind our method on the average of the five benchmarks. This is primarily caused by information loss during vision transformation, which is more prominent in complex visual scenarios. Additionally, object-level visual features also deliver suboptimal performance with 1.9\% lower performance than our method averagely and with additional latency, as object-level visual features are semantically irrelevant and fail to relay accurate visual information.
>
>
> | | CHAIR (Perf↓, Latency↓) | POPE (Perf↑, Latency↓) | AMBER (Perf↑, Latency↓) | MMHal-Bench (Perf↑, Latency↓) | HallBench (Perf↑, Latency↓) |
> |--|--|--|--|--|--|
> | LLaVA-NeXT-7B | 43.0 / **3.16** | 91.0 / **2.46** | 89.4 / **2.79** | 47.9 / **3.48** | **53.1** / 3.91|
> | Compressed Visual Features (Pooling) | 43.9 / 3.33 | 89.6 / 2.61| 89.2 / 2.95 | 42.5 / 3.64 | 46.7 / 4.05 |
> | Compressed Visual Features (MLP) | 42.7 / 3.40 | 91.6 / 2.73  | 90.4 / 3.02 | 44.9 / 3.77 | 51.8 / 4.18 |
> | Compressed Visual Features (Transformer) | 42.2 / 3.61 | 91.9 / 2.96            | 91.1 / 3.27 | 48.5 / 3.99 | 53.3 / 4.40 |
> | Object-level Visual Features    | 42.5 / - | 92.3 / - | 91.8 / - | 47.2 / - | 53.4 / - |
> | **Unimodal Tokens (Ours)** | **41.2** / 3.47 | **93.3** / 2.79|  **92.7** / 3.10 | **51.1** / 3.83 | **55.7** / 4.23|
> \* Object-level visual features are segmented by external model, thus cannot calculate end-to-end latency
>
> >**5. Correction Capability in Adversarial Scenarios**:
>
> To assess the performance on noisy or adversarial scenarios and the correction capacity, we compare our ViF with the baseline under the condition of inputting edited (randomly masked) or mismatched (from other questions) images across multi-agent turns. Specifically, we randomly inject the adversarial images across the first five turns, and evaluate their performance in the following turns (Agent Turn 5, 10, 15, 20). As shown **below** (complete in **Figure 10**), ViF outperforms the baseline by 2.7\% and 5.1\%, respectively. It illustrates that base VLMs in multi-agent settings typically over-rely on prior texts to relay erroneous visual information, which rigidly cling to and continue prior hallucination. Conversely, our proposed visual flow dynamically revises visual cognition across agent turns and alleviates hallucination propagation and snowballing, enabling enhanced correction and anti-adversarial performance.
>
>
> | Inputs | Method | Agent Turn 5 | Agent Turn 10 | Agent Turn 15 | Agent Turn 20 |
> |--|--|--|--|--|--|
> | Edit  Image   | LLaVA-NeXT-7B | 78.7 | 86.9 | 89.2 | 90.9 |
> | Edit  Image   | +Ours | **79.0** | **88.5** | **91.3** | **93.6** |
> | Mismatched Image   | LLaVA-NeXT-7B   | 63.7   | 77.1   | 81.6   | 83.4|
> | Mismatched Image   | +Ours | **64.5** | **80.9** | **85.7** | **88.5** |
>
> ### **Response to Questions**
> >**1. Cross-domain Generalization**:
>
> As described in our response to **Weakness 2**, we assess our ViF on four augmented visual benchmarks, including multi-image and video tasks, with an average improvement of 2.0–4.9\%.
>
> >**2. Further Applications**:
>
> After careful consideration of the two papers and your comments, we believe that models equipped with gated DeltaNet and gated attention (e.g., Qwen3-Next) can still be integrated with our ViF and derive benefits from it. Although gated DeltaNet does not explicitly output attention, attention scores remain accessible in the gated attention module. Our proposed key-norm based strategy is also compatible with the gated structure. Furthermore, from a theoretical perspective, models with a gated structure effectively alleviate the attention sink issue, which reduces unnecessary attention allocation to unimportant tokens. However, attention to partial vision tokens that convey significant visual information remains intact and is even enhanced, ensuring the feasibility of the procedure of ViF.

---

> > ### Comment · Reviewer_HX6A · 2025-11-22
> >
> > Thank you for your response. The rebuttal fully addresses all of my concerns. Therefore, I decide to raise my score accordingly.

---

### Official Review · Reviewer_6R98 · 2025-11-01

**Soundness:** 3
**Presentation:** 3
**Contribution:** 3
**Rating:** 6
**Confidence:** 3

**Summary:**

This paper introduces a new task setting for VLM multi-agent system (MAS), hallucination snowballing. Hallucinated textual response from early agent turns may get passed down to subsequent turns, causing agents to receive spurious textual input and thus inducing further errors and hallucinations. Through extensive analysis of attention distribution over different input token segments, agent turns and transformer layers, the authors discovered that unimodal visual tokens in middle layers produced the most accuracy change when these tokens are removed. As such, unimodal tokens would potentially be most impactful for visual text interaction and is selected as visual relays tokens to carry visual information to subsequent turns. With this finding, the authors propose ViF, which concatenates these relay tokens with instruction and visual tokens to enhance downstream agent inference. Additionally, an attention reallocation module is designed to shift inactive attention to other regions and promote formation of unimodal token patterns. ViF method is extensively evaluated across 8 benchmarks on 6 different base models. Experimental results surpassed previous methods and proved the effectiveness of ViF.

**Strengths:**

The authors introduce a new task setting for MAS hallucination and create a multi-turn evaluation metric which could be beneficial for future researchers.

This paper presents a rigorous evaluation of ViF, testing its generality across six base models of varying sizes and architectures. The authors also provide extensive ablation studies and parameter analysis to validate the method and demonstrate the impact of its components and parameter choices.

The authors also work to improve the method's speed and efficiency by maintaining compatibility with Flash Attention modules. Instead of using attention scores, alternative token selection methods based on key norms and value norms are tested and presented in Table 7 and Figure 8. These experiments support the validity of using key norms as a proxy for attention scores, thereby maintaining compatibility with Flash Attention.

**Weaknesses:**

While the paper is well-written with a clear logical flow, it omits crucial experimental details regarding the MAS evaluation. Specifically, it does not explain how conventional single VLM evaluation benchmarks are adapted for the MAS setting. This omission may hinder readers unfamiliar with multi-agent setups from fully understanding the significance of evaluation results with respect to snowballing hallucination. The paper would be strengthened by providing more prerequisite information on MAS evaluation protocols.

**Questions:**

1. Table 2 shows experimental results for 4 different MAS structures. How are the conventional single VLM evaluation datasets such as POPE tested in multi-agent settings? Could you please provide an example of how a Yes/No question with image is processed in multi-turn agent turns?
2. Could you please provide more experimental setup details for comparison results in table 5? How are other SOTA methods evaluated in MAS settings following circular MAS structure?
3. There is a potential inconsistency in classifying ViF as plug-and-play. The method applies a trained transformer block to embed visual relay tokens. If this transformer block must be retained for each new base model (particularly one with a novel visual encoder), it would constitute a model-specific training phase, which is contrary to the concept of a true plug-and-play component.
4. The paper proposes a new HS metric for snowballing hallucination. The severity of hallucination score, h, is said to be derived from a judge. Could you please provide more information about this judge and how to obtain h from the 8 benchmarks?
5. As shown in table 1, dropping rise and fall tokens in middle layers also contributed significant accuracy change, indicating that they are also important (though not the most important) in model inferencing. Would it be possible to consider combining multiple types of visual tokens as relay tokens? How would the model perform when selecting a portion of rise, fall and unimodal tokens as relay tokens?

---

> ### Author Response · Authors · 2025-11-21
> **Response to Reviewer 6R98 [1/1]**
>
> We greatly appreciate your insightful and detailed feedback. Your comments have been carefully discussed and fully taken into account, and we improve our work accordingly.
>
>
> ### **Addressing Weaknesses**
> >**1. Supplement to MAS**:
>
> Thank you for pointing out the potential room for improvement of our work. Apart from the existing experimental details, we further add a subsection (**Appendix C.1**) that elaborates on the construction, structure, and workflow of the MAS in detail. It specifically illustrates how the final outputs are generated in multi-agent environments. Additionally, as shown in **Figure 12**, we provide four examples to more intuitively demonstrate this working process. We hope these efforts may improve your comprehensibility of the article and ease your concerns.
>
>
> To summarize, the basic MAS environments utilized in this work  mainly follow the [R1]. It adopts a multi-turn interaction system: across network edges, adjacent actors are assigned to nodes and critics to edges in iterative interactions. The prior actor requests feedback, the critic provides reflective suggestions for refinement, and the subsequent actor generates an optimized artifact, iteratively enhancing the output. This framework extends single-agent paradigms to multi-agent settings, reduces context length growth from quadratic to linear, and enables collaborative scaling laws in MAS. Thus, we follow this paradigm to construct basic MAS.
>
>
> [R1] Chen Qian, et al. Scaling large language model-based multi-agent collaboration. In The Thirteenth International Conference on Learning Representations (ICLR), 2025
>
>
> ### **Response to Questions**
> >**1. Multi-agent Settings and Examples**:
>
> To explicitly demonstrate the settings and workflow of the MAS structure, we supplement detailed descriptions in **Appendix C.1** and also four examples in **Figure 12**. For yes/no questions in particular, while the ground-truth answer is limited to a binary choice, the actual output may incorporate the rationale of the answer (see **Examples (a) and (b) in Figure 12**).
>
>
> >**2. Experimental Setups in Table 5**:
>
> We add extra explanations of the  experimental setups in Table 5 (now **Table 6** in the latest manuscript). Specifically, we modify the base agent in the MAS by applying the hallucination mitigation strategies of other SOTA methods. All other settings, particularly the multi-agent setups, remain unchanged, aiming to evaluate the effectiveness of these strategies in mitigating multi-agent hallucination snowballing.
>
>
> >**3. Potential Inconsistency of "Plug-and-play"**:
>
> Thank you for pointing out potential oversight in this regard. Although our ViF requires additional training for different base models, its effectiveness across various models has been proven, confirming its model-agnostic property. Thus, we correct the relevant statement from "plug-and-play" to "model-agnostic" to reduce potential misunderstanding.
>
>
> >**4. Supplement to HS Metric**:
>
> Thank you for pointing out potential ambiguity. Apart from the calculation process of the HS metric, we now include the version and scoring prompt of the judge model employed to calculate the hallucination score. Furthermore, we validate the robustness of the judge by comparing it with another judge model.
>
>
> >**5. Comparison with Combinations of Vision Tokens**:
>
> We sincerely understand and appreciate your suggestion. Based on what we discussed in the requisite analyses in **Section 2**, we believe that unimodal vision tokens are a more efficient and effective medium for relaying visual information than other types of vision tokens. To further prove this point, we include three vision token subset combinations as visual relay tokens: 1) Unimodal + Rise, 2) Unimodal + Fall, and 3) Unimodal + Rise + Fall. Precisely, we randomly select tokens in the another two subsets, and remain the dominant quantity of unimodal tokens. As listed **below** (complete in **Table 11**), these combinations offer no notable performance gains but increase inference time. We nonetheless recognize that more optimal token subsets might be achieved via more rational combinations, not mere random selection and direct addition. Additionally, we also compare compressed and object-level vision features, validating the superiority of ViF.
>
> | Visual Flow | CHAIR (Perf↓, Latency↓) | POPE (Perf↑, Latency↓) | AMBER (Perf↑, Latency↓) | MMHal-Bench (Perf↑, Latency↓) | HallBench (Perf↑, Latency↓) |
> |--|--|--|--|--|--|
> | (e) Unimodal + (a) Rise | **41.0** / 3.55 | 92.7 / 2.87 | 92.1 / 3.20 | 50.6 / 3.87 | 54.8 / 4.33 |
> | (e) Unimodal + (b) Fall | 42.0 / 3.56 | 92.3 / 2.89 | 91.3 / 3.22 | 49.7 / 3.88 | 53.7 / 4.34 |
> | (e) Unimodal + (a) Rise + (b) Fall | 41.3 / 3.63 | 93.1 / 2.94 | **92.9** / 3.28 | 50.3 / 3.94 | 54.1 / 4.40 |
> | **(e) Unimodal (Ours)** | 41.2 / **3.47** | **93.3** / **2.79** | 92.7 / **3.10** | **51.1** / **3.83** | **55.7** / **4.23** |

---

> > ### Comment · Reviewer_6R98 · 2025-11-24
> > **Response to rebuttal**
> >
> > I thank the authors for addressing my questions. My concerns have been adequately addressed, and I am raising my score to 8.

---

### Author Response · Authors · 2025-11-21
**General Responses**

We highly value the constructive and meaningful feedback from all reviewers. Having carefully considered their comments, we revise our submission accordingly in the latest version. To summarize, we conduct following supplementary experiments:


**1. Cross-domain Evaluation (Table 4)**: we assess our ViF on augmented visual benchmarks, including multi-image, video based tasks, demonstrating great applicability and generalization.


**2. Latency and Computational Overload (Table 8,15)**: We quantitatively evaluate inference latency and computational (FLOPs) overloads, illustrating that our ViF brings limited additional costs. It also remains efficient in terms of different model scales, image resolutions and agent turns.


**3. Comparisons with Transformed Visual Features/Combinations of Tokens (Table 11)**: We compare our selected vision tokens with other transformed visual features to relay visual information among agents. Compared with compressed/objective-level visual features as well as different combinations of vision tokens, our ViF exhibits superiority, efficiency, and robustness.


**4. Performance on Adversarial Visual Inputs (Figure 10)**: We input edited and mismatched images and assess the performance in following agent turns, demonstrating superior correction capability over baselines.


**5. Combination with Other Hallucination Mitigation Strategy (Figure 11)**: We combine our ViF with partial existing hallucination mitigation tailored for single model, yielding additive improvements.


Additionally, we provide case study, additional experimental details and analyses, and explanations of of relevant content, improving the clarity of our method and reducing potential confusion. The modifications are colored in blue in the latest version of manuscript. We sincerely hope our responses have effectively addressed the concerns raised by the reviewers and clarified any potential confusion, and we are happy to engage in further discussions if needed.

---

### Meta-Review · Area_Chair_Kgzc · 2025-12-26

**Summary:**

All reviewers found some merit in this work, and overall the paper received positive reviews.

**Reviewer Concerns:**

Most of the concerns were addressed in the rebuttal.

**Reviewer Scores:**

Reviewers would likely have raised their scores after reviewing the rebuttal.

---

### Decision · Program_Chairs · 2026-01-26

Accept (Poster)